# A general rule on the organization of biodiversity in Earth's biogeographical regions

R. Bernardo-Madrid [1,2,3,4] ✉, M. González-Suárez [3], M. Rosvall [1], M. Rueda [2], E. Revilla [4], M. Carrete [5], J. L. Tella[4], J. Astigarraga [6,7] & J. Calatayud [8,9]

Life on Earth is a mosaic distributed across biogeographical regions. Their regional species pools have experienced distinct historical and eco-evolutionary pressures, leading to an expected context-dependent organization of biodiversity. Here we identify a general spatial organization within biogeographical regions of terrestrial and marine vertebrates, invertebrates and plants (more than 30,000 species). We detect seven types of areas in these biogeographical regions that reflect unique combinations of four fundamental aspects of biodiversity (species richness, range size, endemicity and biogeographical transitions). These areas form ordered layers from the core to the transition zones of the biogeographical regions, reflecting gradients in the biodiversity aspects, experiencing distinct environmental conditions, and exhibiting taxonomic dissimilarities due to nestedness. These findings suggest this ubiquitous organization is mainly driven by the action of two complementary environmental filters, one acting on species from regional hotspots and the other on species from permeable biogeographical boundaries. The influence of these regional filters extends across spatial scales and shapes global patterns of species richness. Regional biodiversity follows a universal core-to-transition organization governed by general forces operating across the tree of life and space.

Biogeographical regions—or bioregions—reflect regional species pools with different origins, compositions and ecological and evolutionary characteristics[1-6]. These regional species pools are, in part, isolated by geological and climatic barriers[7-9] and have experienced distinct historical, ecological and evolutionary pressures[9-13]. Therefore, it is reasonable to expect a context-dependent organization of regional biodiversity with idiosyncratic variations across biogeographical regions and taxa. However, the existence of global factors, processes and biodiversity patterns that transcend biogeographical region boundaries[14,15], or tend to be consistent across regions[16,17], open the door for general processes and mechanisms to exert a stronger influence than the idiosyncrasies of geographical areas and life forms; forcing regional biodiversity to organize into limited, convergent and more predictable ways. These unevaluated alternatives revolve around the predominance of context-dependent versus general processes and mechanisms, offering contrasting perspectives on the principal forces driving the organization of biodiversity across multiple scales and, thus, on our understanding of the distribution of life on Earth.

Species biodiversity can be described using complementary aspects. For instance, geographical areas: may teem with species or harbour only a few[18] (species richness); may contain a mixture of biotas from different biogeographical regions, as observed in transitional areas, or maintain mainly species from a single biogeographical region[8] (biota overlap); may host species occupying relatively small or large

areas within the biogeographical region[19] (species range size or occupancy); and may predominantly host endemic or non-endemic species[19] (species endemicity). These four complementary aspects—species richness, biota overlap, species occupancy and species endemicity—play pivotal roles in ecology and biogeography[9,18,20–22]. Hence, distinct combinations of values of these complementary aspects and their spatial distributions in the biogeographical regions can be understood as the biogeographical organization of biodiversity. The study of such organization can offer new insights into how regional biodiversity is assembled and what its drivers may be. Moreover, by comparing the organization of biodiversity across biogeographical regions and taxa, we can determine whether biodiversity organization and its drivers are context-dependent or general.

To address this, we assessed the joint spatial patterns of the four complementary biodiversity aspects in biogeographical regions of seven ecologically contrasting taxa: amphibians, non-marine birds, dragonflies, non-volant mammals, rays, reptiles and trees. Our results reveal a general pattern in the organization of regional biodiversity across global biogeographical regions of terrestrial and marine vertebrates, invertebrates and plants. In this study, we describe this regional spatial pattern across the tree of life and space, provide empirical evidence for its underlying mechanisms, and show how these regionally operating processes influence species richness patterns at the global scale.

## Life forms and data studied

We studied the biogeographical regions of seven contrasting life forms using global distribution maps for five taxa: amphibians (6,563 species), non-marine birds (9,752 species), non-volant terrestrial mammals (4,200 species), reptiles (8,219 species) and rays (360 species). We also used distribution maps for dragonflies in Eurasia (648 species) and forest inventory data for trees in North America (307 species). These more than 30,000 species occupy distinct habitats (marine and terrestrial), and have distinct mobilities (sessile, aquatic, aerial and terrestrial), life histories (slow or fast) and physiologies (ectotherm or endotherm). Furthermore, the dataset encompasses different extent, resolution and collection methodologies. Thus, obtaining consistent results would support the generality of our findings to variations in life forms and data characteristics. To delineate biogeographical regions and identify their most characteristic species so as to estimate their biodiversity aspects, we projected species distribution data (distribution maps or inventory data) onto a distinct regular grid for each taxonomic group. For each taxon, we built a bipartite network, linking species to the grid cells where they occur (Extended Data Fig. 1), and applied the widely used community detection algorithm Infomap[23,24]. This algorithm simultaneously identifies modules of highly connected grid cells and species in an integrated approach[19,25]. Grid cells within a module define a biogeographical region[19] (Extended Data Figs. 2–8). Species assigned to the same module are considered characteristic of that biogeographical region[19], meaning their distribution range is largely confined to that region, contributing to its unique biotic identity. By contrast, species present in a bioregion's grid cells but not grouped in the same module are deemed non-characteristic[19], either because they are more strongly associated with another region or display an even distribution that precludes a clear regional association (see graphical description in Extended Data Fig. 1).

## General organization of biodiversity

In each grid cell representing regular geographical areas on Earth, we measured four complementary biodiversity aspects: the ratio of characteristic and non-characteristic species, as well as the richness, occupancy and endemicity of characteristic species (hereafter, biota overlap, species richness, occupancy and endemicity) (Supplementary Figs. 1–7 and Supplementary Tables 1–8). These four aspects were used in a single *k*-means clustering analysis to group grid cells with similar

combinations of biodiversity values across biogeographical regions and taxa (see workflow in Extended Data Fig. 1). This *k*-means clustering included all grid cells from the seven taxa, treating each grid cell as a distinct analytical unit, regardless of its grid membership. The resulting clusters represent geographical areas (groups of grid cells) with similar combinations of biodiversity values across biogeographical regions and taxa, which we refer to as 'biogeographical sectors'. In theory, the potential number of biogeographical sectors could be large. For instance, even if the four studied biodiversity aspects only had three values each—low, medium and high—there would be 81 potential value combinations ($3^4$). If biodiversity is organized differently across the taxa, grid cells for each of the seven studied life forms would be clustered separately from each other. Conversely, if biodiversity is organized similarly across the taxa, clusters would combine grid cells from all the taxa. Our *k*-means clustering showed an optimal number of seven biogeographical sectors, each encompassing grid cells from all taxa (Fig. 1a). The generality and low number of biogeographical sectors across the seven taxa support the hypothesis that biodiversity at regional scales is arranged in a consistent and limited number of ways, and that the mechanisms governing regional biodiversity transcend the particularities of individual life forms.

To interpret the biogeographical meaning of the identified sectors, we analysed the distribution of biodiversity aspects across them using values from all the grid cells of the seven taxa (Fig. 1a). We also assessed the similarity of sectors in a multidimensional space defined by four biodiversity aspects using principal component analysis (Fig. 1b). Finally, we mapped sectors and, for each taxon and biogeographical region, assessed whether neighbouring relationships between sector pairs occurred more frequently than expected by chance (one-sided binomial proportion tests, expected probability = 1 in 6, $P < 0.05$) (Fig. 1c and Supplementary Table 9). Our results show that the biogeographical sectors exhibit a spatial neighbouring pattern with an ordered layered scheme (Figs. 1–3). Shifts between neighbouring sectors reflected spatial gradients in two or more of the biodiversity metrics (Fig. 3a,b). On one hand, the ordered layered scheme of the biogeographical sectors largely captured two opposing gradients of variation in regional species richness and the overlap of biotas (Figs. 1 and 2). At one extreme of these gradients lie the regional species hotspots, likely areas with favourable conditions for the diversification and persistence of most regional species[26]. At the other extreme are the most transitional areas located near the permeable boundaries of biogeographical regions. The sectors between these extremes partially reflect the geographical proximity of areas to those regional hotspots[27] and the permeable boundaries of biogeographical regions[9,28]. On the other hand, the biogeographical sectors also mirrored two inverse gradients involving species occupancy and endemicity, with hotspots harbouring the most endemic and least widespread species, whereas the permeable boundaries harboured the least endemic and most widespread species (Fig. 1a). Thus, biodiversity in biogeographical regions generally aligns with what we term a 'core-to-transition' organization.

The biogeographical sectors comprising the sequence of ordered layers can vary across biogeographical regions, probably reflecting the challenges in discretizing what may be a continuum[10] (Fig. 3c). For instance, some regions only contain sectors with a high overlap of biotas, probably reflecting the widely recognized transitional nature of some biogeographical areas[9,29,30]. Examples include the Mexican, South American, Saharo–Arabian and Oriental transitional zones (Figs. 1d–g and 2). Similarly, the sequence of layers can start from different points, such as the centre or boundaries of biogeographical regions, or vary in their orientation, ordering latitudinally or longitudinally, probably reflecting the idiosyncrasies of geographical areas and biotas (Figs. 1d–g and 2). However, the biogeographical meaning of all the observed sequences of the layers prevails, supporting the generality of the conceptual core-to-transition organization (Fig. 3). Sensitivity analyses using *k*-means clustering with 2 to 8 clusters (general biogeographical

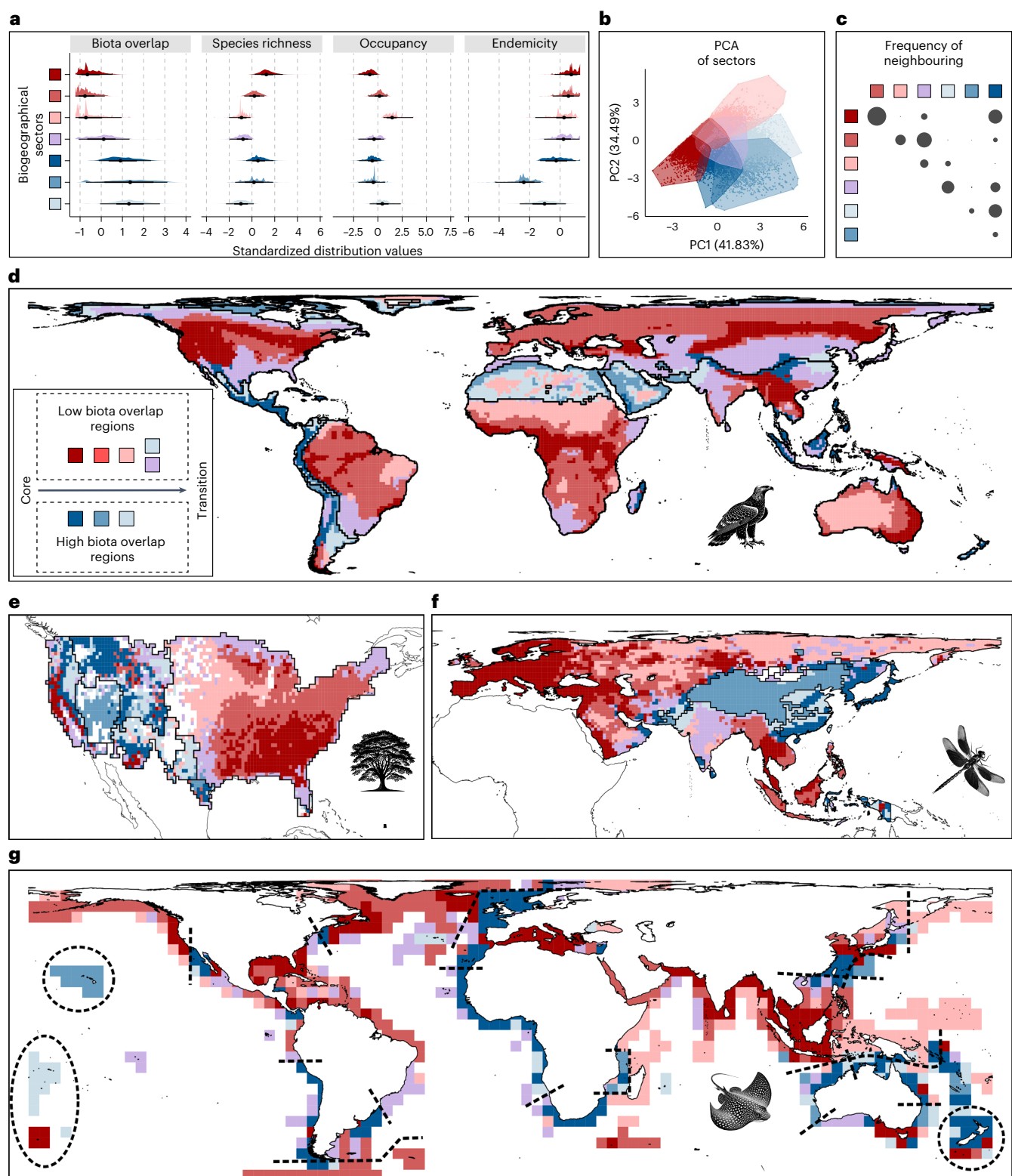

**Fig. 1 | Seven general and spatially structured biogeographical sectors characterized from four biodiversity aspects across taxa. a**, Distribution of biodiversity values from 48,870 cell–taxon combinations across sectors (represented by colours, see **d**). Black dots denote the median and the thick and thin lines indicate the 66% and 95% quantile intervals, respectively. **b**, Two main axes from a principal component analysis (PCA) of the 48,870 cell–taxon combinations. **c**, Relative pairwise frequency of higher-than-expected neighbouring between sectors across regions and taxa (one-sided binomial proportion tests under a null expectation of 1 in 6; $P < 0.05$). Circle size is proportional to the frequency of neighbouring. A total of 1,308 sector pairs were evaluated. The observations and relative frequencies per pair are reported in Supplementary Table 9. **d**–**g**, Spatial distribution of sectors in bioregions of birds (**d**), trees (**e**), dragonflies (**f**) and rays (**g**). Black lines delineate bioregion boundaries. Red and blue colours tend to indicate bioregions with overall low and high biota overlap, respectively. Darker tones represent sectors of high richness and endemism, with lighter tones indicating sectors dominated by widespread species. Icons indicate the taxonomic group to which bioregions correspond.

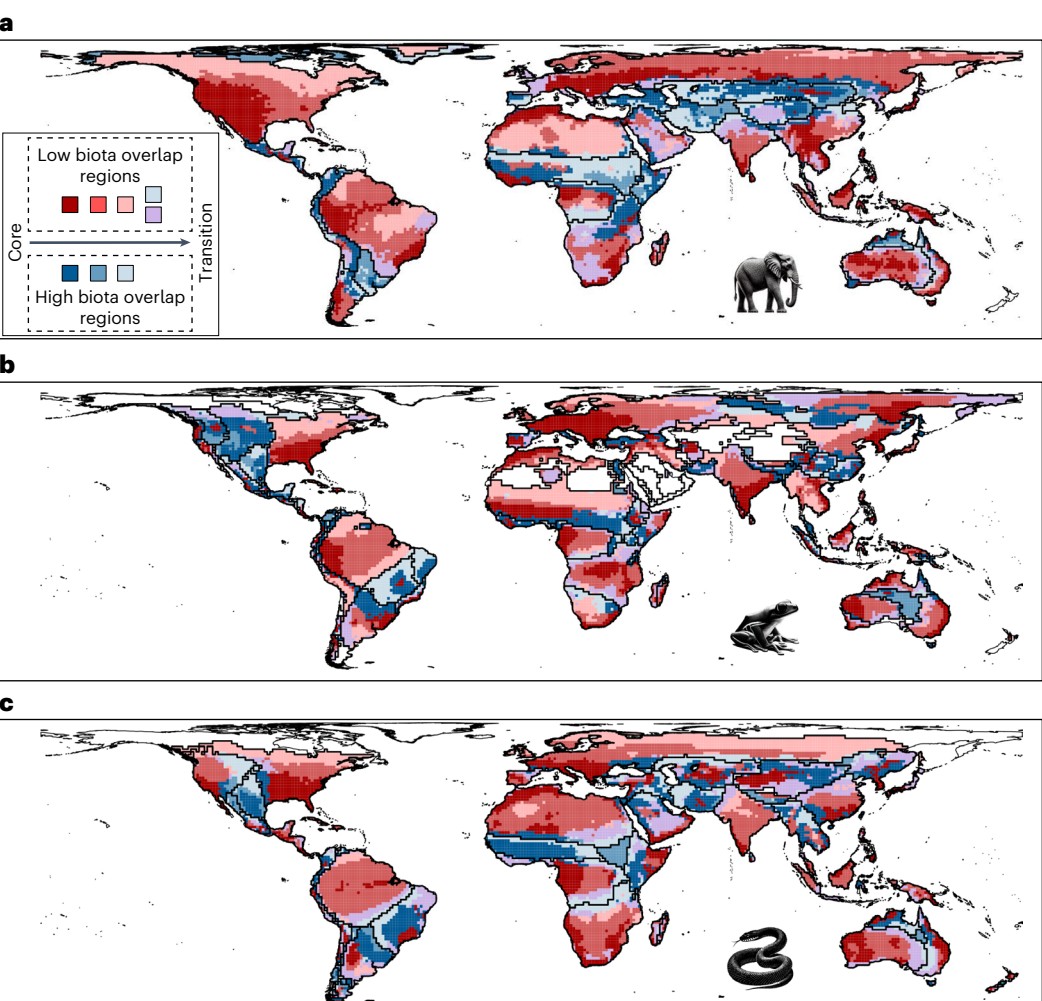

**Fig. 2 | Seven general and spatially structured biogeographical sectors characterized from four biodiversity aspects across diverse taxa. a–c,** Spatial distribution of the biogeographical sectors in bioregions of mammals (**a**), amphibians (**b**) and reptiles (**c**). Icons indicate the taxonomic group to which bioregions correspond. The biogeographical interpretations of the sectors and colours are provided in the legend of Fig. 1.

sectors) also show the core-to-transition organization and the gradient in the four biodiversity aspects (Supplementary Figs. 8–13).

## Underlying mechanism

Which mechanism could account for this general organization of biodiversity across biogeographical regions in different parts of the world and across life forms? Gradients in species richness[31], range size[32] and transitional zones[9] have been associated with environmental filters[31,33]. Thus, hypothetically, the observed decreasing richness and increasing occupancy covariation patterns might reflect that only a reduced subset of tolerant species can colonize some areas while expanding their distribution ranges[1,34]. Similarly, a decreased overlap of biotas from the boundaries can result from a filtering of species from other biogeographical regions[1,35]. If biogeographical sectors reflect environmental filtering from regional hotspots and permeable barriers, we would expect the biogeographical sectors in a biogeographical region to be associated with distinct environmental conditions. Additionally, differences in species compositions among these biogeographical sectors would predominantly arise from one sector's species being a subset of those present in another, representing nestedness patterns rather than species turnover[36,37].

In line with our expectations, multinomial logistic regressions for each biogeographical region and taxon showed that the biogeographical sectors occupy areas with distinct environmental conditions in

97.7% of the cases (median McFadden's pseudo-$R^2$ in multinomial logit models across taxa = 0.32 (Fig. 4a) using temperature and precipitation as explanatory variables in the terrestrial taxa and temperature and salinity at the sea surface in the marine taxon (Supplementary Figs. 14–20)). Complementarily, the partitioning of species dissimilarity into nestedness and turnover components[38] across biogeographical sectors within a given biogeographical region revealed that taxonomic dissimilarity is more attributed to nestedness in 77 ± 2% of the biogeographical regions across all taxa (mean ± standard error (s.e.) of the proportion of biogeographical regions with higher nestedness than turnover across all taxa (Fig. 4b)). In some cases, our environmental variables exhibited limited explanatory power in explaining the distinct biogeographical sectors, probably due to their association with other non-studied environmental, historical and geographical factors[1,36,39]. For example, changes in temperature and precipitation since the Last Glacial Maximum also correlate with the core-to-transition pattern (sensitivity analysis) (Supplementary Information Appendix A and Supplementary Figs. 21–26). Similarly, in certain cases, species turnover was higher than nestedness, probably due to the intricate overlap of biogeographical processes and biotas across spatio-temporal scales[2,40] as well as to changes in biogeographical patterns resulting from human-mediated extinctions and species introductions[19,41]. However, most of the results across the biogeographical regions and taxa aligned with our predictions about the different environmental

the colonization of species from other biogeographical regions seems to have also been constrained by environmental factors[9,28,34], producing the observed gradient in the biota overlap from permeable borders. Thus, regional biodiversity could be largely conceptualized as species

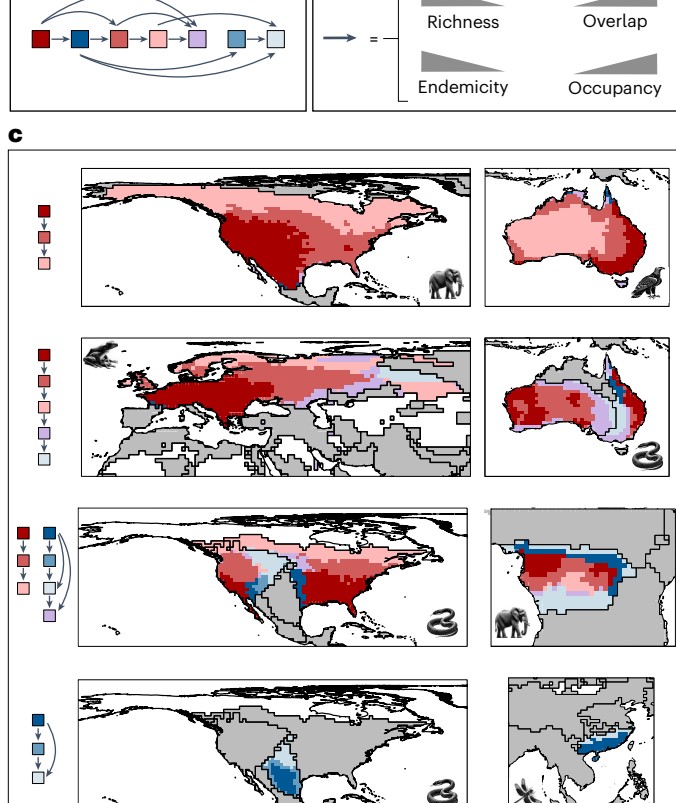

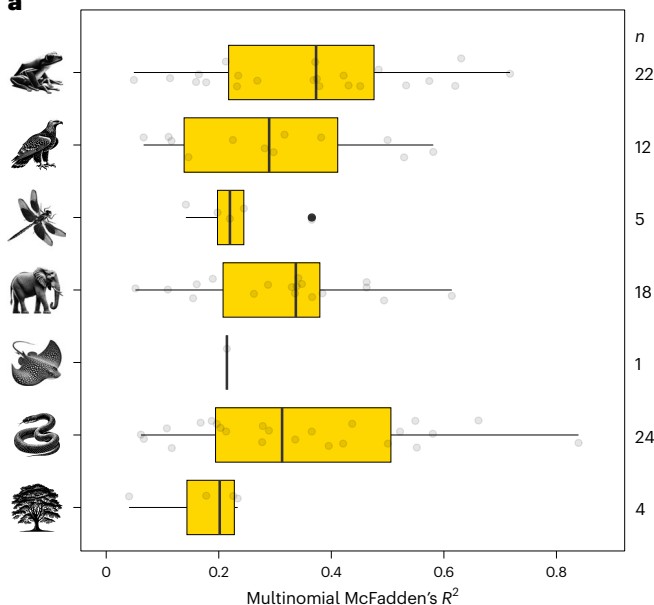

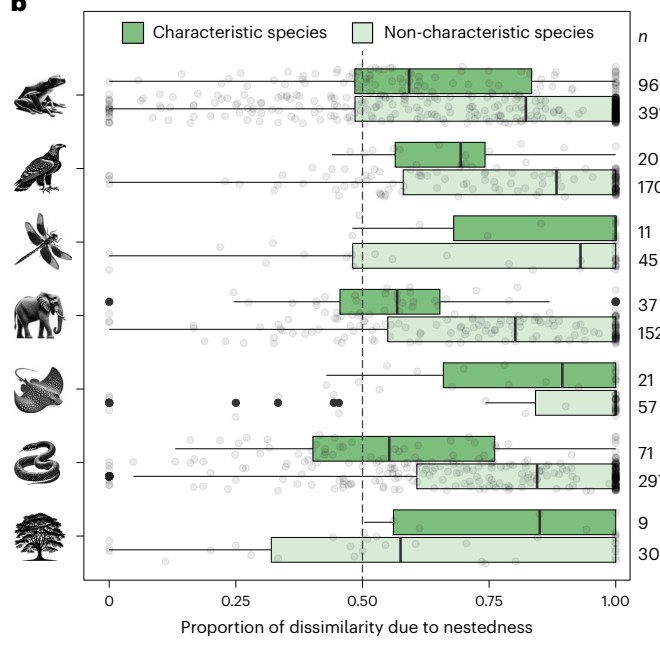

**Fig. 3 | Core-to-transition sequence of biogeographical sectors. a**, Illustration of the ordered sequence observed in the sectors, with links indicating statistically significant neighbouring probabilities (see also Fig. 1b). **b**, Gradients in the four biodiversity aspects represented by the links in **a**. **c**, Examples of areas showing alternative sequences of neighbouring sectors across biogeographical regions and taxa, but with similar biogeographical meaning: adjacent sectors from cores to transition areas tend to show simultaneously lower richness and higher overlap of biotas, as well as be occupied by species with larger occupancies and lower endemicities.

conditions associated with distinct biogeographical sectors, with nestedness being the key component of their biotic dissimilarity. The results from the multinomial and nestedness analyses remained consistent when varying the number of biogeographical sectors from seven to 2–8 (sensitivity analyses) (Supplementary Figs. 8–14 and Supplementary Tables 10 and 11). The congruency between our predictions and our results supports the important and general role of environmental filters acting on both characteristic and non-characteristic species in shaping biodiversity within biogeographical regions.

Taken together, these findings support the hypothesis that regional hotspots, probably representing centres of diversification or past climatic refugia[42,43], act as species sources from which species with better dispersive capabilities[27] and greater environmental tolerances would have colonized other areas in the biogeographical region, expanding their ranges[34,35]. This hypothesis[27] is supported by the fact that the most core areas of each biogeographical region cover approximately 30% of the region's surface, but harbour more species than the remaining 70% of the area (holding around 90% of the species, a species richness higher than expected by chance, based on the geographical extent of biogeographical sectors for reptiles, amphibians, dragonflies and trees; $P < 0.05$ in models permuting a sector's identity). Similarly,

**Fig. 4 | Biogeographical sectors are consistent with environmental filtering showing associations with environmental variables and community dissimilarity attributed to nestedness. a**, McFadden's pseudo-$R^2$ from multinomial logistic models across combinations of biogeographical regions and taxa, using biogeographical sectors as the dependent variable and environmental factors as the explanatory variables. See model results across bioregions and taxa in Supplementary Figs. 14–20. **b**, Proportion of dissimilarity due to nestedness among assemblages of characteristic (dark green boxplots) and non-characteristic (light green boxplots) species across sectors within the same biogeographical region. In both panels, circles represent individual analyses conducted for each biogeographical region. In the nestedness analyses, we conducted one analysis for each origin of the non-characteristic species. On the boxplots, the solid line is the median, the black box is the interquartile range, the whiskers are the minimum and maximum values excluding outliers and the circles are the outliers.

sorting from two predominant sources, entailing dispersion from the most representative or suitable areas in the biogeographical region and the colonization of biotas from other biogeographical regions.

## Influence across spatial scales

The ubiquity of the core-to-transition organization of biodiversity worldwide, in conjunction with the interplay of biogeographical processes across spatial scales, raises a new question: can the core-to-transition organization help us understand global variations in local species richness?—a long-debated topic in ecology and biogeography[18,39,44]. Variations in species richness at local scales, such as our grid cells, are expected to be regionally determined by the number of species in the biogeographical region—that is, the size of the regional species pool resulting from the balance of speciation, extinction and biogeographical dispersal[45,46]—and by the sorting of the regional species pool across the biogeographical regions, largely influenced by environmental filters[47,48] and associated here with the core-to-transition organization. However, the relative importance of these two drivers remains uncertain. If the size of the regional species pool primarily explains the variance in species richness, then global patterns may be predominantly influenced by biogeographical isolation and connectivity behind the formation of biogeographical regions, as well as the context-dependent speciation and extinction events within them. Conversely, if the sorting of characteristic and non-characteristic species better explains the local variation in species richness, this may underscore the predominant role of regional environmental filters across the planet.

To address the relative importance of core-to-transition organization in shaping global patterns of species richness, we modelled the variance in species richness across grid cells using three variables: the size of the regional species pool, the variations due to the sorting of characteristic species and the variations due to the sorting of non-characteristic species. We measured the size of the regional species pool as the total number of characteristic species in each bioregion, while the sorting of characteristic and non-characteristic species was measured as the observed richness of characteristic and non-characteristic species minus their respective mean values in the biogeographical regions—that is, centred values to discount for the effects of the regional species pool size[46]. Note that the size of the species pool plus the centred species richness of the characteristic and non-characteristic species can be viewed as an approximated decomposition of the observed species richness allowing us to explore the independent effect of each component on the local species richness (Methods). Linear regressions and variance partitioning showed that the influence of species sorting can be comparable to that of the regional species pool size in some cases (dragonflies and mammals) or even greater in others (rays) (mean ± s.e. of the non-shared variance explained by the sorting of characteristic species = 0.25 ± 0.06, the sorting of non-characteristic species = 0.06 ± 0.01 and the size of the regional species pool = 0.39 ± 0.06) (Fig. 5). Thus, the processes underlying species sorting in biogeographical regions, probably tied to regional environmental filters and responsible for the core-to-transition organization, may be, on average, as important as those variations in the regional pool size driven by the balance of speciation, extinction and biogeographical dispersal. These results advocate for broadening our attention beyond the traditionally evaluated size of the regional species pool when exploring how regional effects drive global biodiversity[46] and, in particular, to also consider the processes and mechanisms driving the core-to-transition organization.

## Conclusions

Species biodiversity in biogeographical regions tends to be spatially sorted into a core-to-transition organization. This finding aligns with a concept that has been implicit in the minds of biogeographers for centuries[5,9,28,29,49]. The generality of the core-to-transition organization

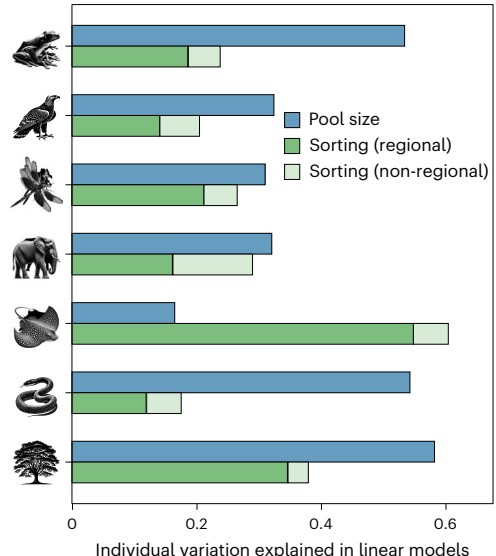

**Fig. 5 | Species sorting within biogeographical regions largely explains local variations in species richness across the planet.** Independent (non-shared) variance explained by factors—sorting and pool size—associated with distinct biogeographical hypotheses. Variance partitioning derived from linear regressions predicting species richness at the grid-cell level with three explanatory variables: the size of the regional species pool (blue bar), the richness of characteristic species (dark green bar) and the richness of non-characteristic species (light green bar).

across biogeographical regions with distinct origins, ages, conditions and histories, as well as of taxa with distinct eco-evolutionary characteristics and requirements, suggests that the organization of regional biodiversity tends to converge in a predictable way. The cumulative evidence across space and taxa suggests the action of general mechanisms, which seem to be related to species filtering from regional hotspots and from the permeable boundaries of biogeographical regions. These findings and hypotheses align with established theory on how species are assembled locally from hypothetical species sources[45,47]. These source areas, which occupy a relatively small area on Earth, probably have an invaluable influence on the biodiversity of the entire biogeographical region, making them potential targets for international conservation[50]. Furthermore, our core-to-transition hypothesis and results show that global variations in species richness can be better understood by unravelling the genesis of regional hotspots and the subsequent filtering of species to the rest of the biogeographical region. In conclusion, this apparent rule on the organization of biodiversity in biogeographical regions, coupled with its relevance for understanding global variations in species richness, supports the processes and mechanisms underlying the core-to-transition organization reflecting some fundamental principles governing life on Earth.

## Methods

### Delineation of biogeographical regions

We obtained the species distributions of amphibians, dragonflies, mammals, rays and reptiles from the International Union for Conservation of Nature's Red List (www.redlist.es), of birds from BirdLife (www.birdlife.org) and of trees from the Forest Inventory and Analyses National Program of the United States (www.fia.fs.usda.gov). We projected the species occurrences of the seven taxa onto seven distinct regular grids, with a resolution of 111 × 111 km for terrestrial animals[19], 55.5 × 55.5 km for trees[51] and 444 × 444 km rays for similarity to previous studies[52]. The number of grid cells varied across each taxon, depending on the species distribution and grid resolution (amphibians = 8,907 grid cells, birds = 10,757, mammals = 10,744, reptiles = 9,507,

dragonflies = 5,110, trees = 3,019 and rays = 826, with the total = 48,870 grid cells). Although grid cells from different taxa could overlap geographically, each taxon was assigned its own distinct set of grid cells. All 48,870 grid cells were treated as independent analytical units in the subsequent $k$-means clustering analyses.

To calculate the biodiversity aspects in the grid cells, such as biota overlap, it was essential to delineate the biogeographical regions and assign species to the biogeographical region with which they were most closely associated. We used a well-established biogeographical method that both delineates the regions and assigns species to them in a single integrated process. Specifically, we employed Infomap[24,53], a community detection algorithm based on network theory[19,24,25,53,54] (www.mapequation.org/infomap/; see details below). In network approaches, the species and grid cells are treated as two types of nodes in a bipartite network, linked based on species occurrence[19,54] (Extended Data Fig. 1). Infomap identifies modules that represent groups of highly connected grid cells and species. Infomap is based on information theory, and these modules represent the best compression of the systems' information, capturing the key structural patterns within the network[24,53], which, in our case, was the co-occurrence patterns of species across the globe[19,54]. The grid cells within a module are considered the geographical areas of a biogeographical region, whereas the species within the same module represent its characteristic species pool[19]. Characteristic species have their entire distribution range, or most of it, in their associated biogeographical region (thus including endemic, but also non-endemic, species). Those species present in a grid cell of a biogeographical region but not assigned to the same module were considered as non-characteristic species[19] (Extended Data Fig. 1). Non-characteristic species can be more affined to another biogeographical region—that is, clustered in another module[19]. The absolute and relative occurrences of these characteristic and non-characteristic species allow the measurement of biodiversity aspects (see below). The biogeographical regions detected are congruent with those proposed in previous studies and other methodological approaches[5,6,25,52,54–59] (Extended Data Figs. 2–8).

Identifying patterns in large communities is a hard problem. Infomap, as with most clustering and network community algorithms, identifies solutions using a heuristic procedure[19,23,24,60]. To consider the heuristic search of Infomap and, thus, all possible biogeographical region delineations (avoiding local minima), we conducted 150,000 analyses for each taxon[19], selecting for subsequent analyses the best delineation based on an information-theoretic criterion of Infomap called codelength[19,24,53,54,61].

We selected the community detection algorithm Infomap over alternative algorithms and clustering approaches for several reasons. First, identifying both the bioregions and their characteristic species is essential for calculating biodiversity metrics and evaluating general biodiversity patterns—our primary goal. Infomap integrates the clustering of grid cells and species into a unified methodological framework[19], thereby avoiding additional steps that could introduce methodological complexity or subjectivity. Second, Infomap ensures one-to-one correspondence between regional species pools and biogeographical regions, aligning with the definition of the biogeographical region—Earth's areas identified as distinct due to the presence of different species pools. Third, Infomap is widely used and accepted in the biogeographic community[19,25,53,61], and its regions have even served as benchmarks for new methods of biogeographical delineation[62]. In particular, Infomap produces biogeographical regions that are comparable to those from well-established biogeographical methods, such as agglomerative hierarchical clustering and modularity-based approaches[25,52,59], as well as being comparable to alternative clustering methods used in other fields of science, such as the stochastic block model[63] (Supplementary Information Appendix B). Moreover, Infomap-based biogeographical regions adequately represent the co-occurrence patterns of characteristic species (for example, the spatial congruence between bioregion boundaries and overlapping distribution ranges of characteristic species[25]). Fourth, Infomap inherently determines the optimal number of clusters—here biogeographical regions—during its search process[23,53], eliminating the need for additional threshold-setting steps that could introduce methodological complexity and subjectivity[9,25]. Fifth, Infomap offers additional advantages over other community detection algorithms[24]. For instance, Infomap may be less affected by the resolution limit[24], which refers to the challenges of accurately identifying communities in datasets with high complexity or large volumes of data (see analyses on sensitivity to data extent below). Moreover, its heuristic search yields more stable, and then reliable, solutions[60]. Finally, in methodological studies comparing method performance against ground-truthing, Infomap consistently ranks among the top-performing methods[64–66]. For a non-specialized description of Infomap and its application in biogeography, see refs. 54,61 and the supplementary material of ref. 19. In summary, we chose Infomap because it provides reliable biogeographical regions and also identifies their characteristic species—a critical step in addressing our primary goal of describing biodiversity patterns in biogeographical regions.

## Biogeographical sectors

We identified geographical areas in biogeographical regions that exhibited similar biodiversity values, calling them biogeographical sectors. To characterize these biogeographical sectors, we used four biodiversity aspects that captured how biogeographical regions are the result of processes acting on both characteristic and non-characteristic species. These four biodiversity aspects were: the relative richness of characteristic species, which quantified how well a grid cell represented the characteristic species pool of a biogeographical region compared to other grid cells; the overlap of biotas, which measured the proportion of non-characteristic species in a grid cell; the relative occupancy of characteristic species, which measured the extent to which a characteristic species occupied its biogeographical region in comparison to other characteristic species of its biogeographical region; and the endemicity of characteristic species, which indicated the proportion of a characteristic species' distribution range outside of its associated biogeographical region.

To quantify these four biodiversity aspects, we employed network cartography[67], a widely used approach in multiple fields of science used for gaining insights into the organization and connectivity of elements in complex systems (in our case, species and grid cells). Specifically, we used two measures: the within-module degree ($z$) and the connectivity across modules ($C$) for both grid cells and species. These metrics informed us on the link distribution of nodes inside and outside their respective module[67] (Extended Data Fig. 1). The $z$ measures represented the relative connectivity of a given node within its associated module, measured as a $z$ score ranging from minus infinity to infinity. For a grid cell, $z_{cell}$ indicated the relative species richness of the characteristic species in a grid cell compared to other grid cells in the same biogeographical region. In each module, the grid cell with the highest $z_{cell}$ best represented the characteristic species richness, and vice versa. For the species, $z_{spp}$ denoted the relative occupancy of a species in its associated biogeographical region compared to the other characteristic species. In each module, the characteristic species with the highest $z_{spp}$ occupied the largest area of the biogeographical region, and vice versa. Contrastingly, the connectivity across modules ($C$) measures the proportion of links of a node outside its module, ranging from 0 to 1. For the grid cells, $C_{cell}$ measures the overlap of biotas as the proportion of non-characteristic species present in a given grid cell. Values of 0 indicate the absence of non-characteristic species with zero overlap of biotas, with lower values indicating the overlap of biotas. For example, a value of 0.4 would indicate that 40% of the species present in the grid cell were non-characteristic. For the species, $C_{spp}$ measures the endemicity, quantified as the proportion of

the characteristic species' distribution area that fell within its biogeographical region. A value of 1 indicates that the characteristic species is endemic, with lower values indicating that part of its distributional area is also present in other biogeographical regions. To assign a value of occupancy, $z_{spp}$, and endemicity, $C_{spp}$, to each grid cell, we selected their present characteristic species and calculated the median values of $z_{zpp}$ and $C_{spp}$ (Extended Data Fig. 1). To quantify the four biodiversity aspects per grid cell, we defined:

$$\text{Relative species richness} = (I_c - \bar{I}_{m_c})/\sigma_{I_{m_c}} \qquad (1)$$

where $I_c$ is the number of links of grid cell $c$ to the species in its module, $m$, and $\bar{I}_{m_c}$ and $\sigma_{I_{m_c}}$ are the mean value and the standard deviation, respectively, of $I_c$ over all grid cells in module $m$.

$$\text{Biota overlap} = O_c/L_c, \qquad (2)$$

where $O_c$ is the number of links of grid cell $c$ connecting with species outside its module, and $L_c$ is the total number of links of grid cell $c$, including links inside ($I_c$) and outside ($O_c$) its module.

$$\text{Endemicity} = \text{Mdn}\left(\frac{I_s}{L_s}\right), \qquad (3)$$

where $I_s$ is the number of links of the present characteristic species, $s$, connecting with grid cells inside its module, and $L_s$ is the total number of links of the present characteristic species, $s$, including links inside and outside its module, and Mdn is the median of the values of the present characteristic species in the grid cell.

$$\text{Relative occupancy} = \text{Mdn}\left(\frac{I_s - \bar{I}_{m_s}}{\sigma_{I_{m_s}}}\right), \qquad (4)$$

where $\bar{I}_{m_s}$ and $\sigma_{I_{m_s}}$ are the mean value and the standard deviation, respectively, of $I_s$ over all characteristic species in module $m$. We calculated the four biodiversity metrics in all the grid cells of each of the seven grids associated with the seven taxa.

The four biodiversity aspects did not strongly correlate (|Pearson's coefficient| < 0.7 (ref. 68)), and provided complementary information (see correlation values for each taxon and overall in Supplementary Tables 1–8).

To divide the grid cells into biogeographical sectors, we followed a two-step approach. In the first step, we conducted the following procedure seven times, once per taxon. We selected the grid cells of a given taxon and performed a $k$-means analysis to identify clusters of geographical areas with similar combinations of the four biodiversity aspects. To determine the optimal number of clusters, for each taxon, we used an elbow-like[69] method commonly employed in biogeography[25,70]. We tested cluster numbers ranging from two to 30, and calculated the goodness-of-fit (GoF), defined as the ratio of the between-cluster variance to the total variance. Recognizing the heuristic nature of $k$-means, we repeated each $k$-means clustering 100 times and selected the partition with the highest GoF in each partition with cluster numbers ranging from two to 30. Then, in a piecewise regression, we modelled the best 29 GoF values (dependent variable) with the number of clusters (explanatory variable) to identify the inflection point or point of sharpest decrease[71], by searching for the cluster count that resulted in the lowest residual standard error. This point reflects where adding more clusters no longer substantially increases the GoF, and thus represents an optimal balance between model complexity and clustering quality. We identified this point using a brute force iterative search[72]. At this stage, we obtained seven outcomes, one per taxon, indicating the optimal partition of grid cells to taxon-specific biogeographical sectors (Supplementary Table 12). We used these outcomes in the second step.

In the second step, to address whether the biogeographical sectors were taxon-specific or general across the taxa, we performed a general $k$-means clustering analysis that included all 48,870 grid cells from the seven taxa jointly. We evaluated 13 potential partitions, ranging from two to 14 clusters, with the maximum number constrained by the results from the taxon-specific analyses (first step) and computational feasibility. These clusters, representing geographical areas with similar biodiversity characteristics across all taxa, were termed general biogeographical sectors. To determine the optimal number of clusters, we sought the partition in the general $k$-means that showed the highest similarity to the taxon-specific results. For each of the 13 potential partitions conducted with all grid cells from the seven taxa (2–14 clusters), we separated the grid cells of each taxon to create seven individual sets. For each partition and set, we quantified the similarity in grid-cell grouping between the general $k$-means (second step) and the taxon-specific $k$-means (first step) using adjusted mutual information (AMI), a widely accepted metric for assessing clustering similarity[73]. For each partition, we obtained seven AMI values, one per taxon, which were averaged to provide an overall similarity measure per partition. Our results showed that the partition with seven clusters produced the highest average AMI value. We considered that the seven-cluster partition provided the best description of biodiversity organization across the seven taxa, and used that as the basis for our subsequent analyses. Nevertheless, all subsequent results remained consistent, regardless of the number of clusters (sensitivity analyses using cluster 2–8 counts) (Supplementary Figs. 8–13 and Supplementary Tables 10 and 11). Although we chose the number of clusters that best represented all taxa, on average, the grid cells were free to group based on their biodiversity aspects. If the combinations of biodiversity aspects in the grid cells differed between taxa, the grid cells of each taxon were expected to form separate clusters. Conversely, if the combinations of biodiversity aspects were similar across taxa, the clusters would contain grid cells from all taxa.

All $k$-means clustering analyses, both taxon-specific and general, used the four biodiversity aspects of species richness, biota overlap, occupancy and endemicity as features. Before clustering, these metrics were standardized to account for different value ranges. We excluded those modules with missing values in any of the four metrics, including modules consisting solely of species without a clear biogeographical affinity, modules in which all the grid cells exhibited identical species richness and modules where all the species occupied an identical number of grid cells. These modules represented tiny biogeographical regions with non-biogeographical relevance to our goals.

Sensitivity analyses accounting for the distinct number of grid cells per taxon in the $k$-means clustering provided similar results (Extended Data Figs. 2–8, Supplementary Figs. 27–34 and details in Supplementary Information Appendix C). To maintain consistency in the $k$-means analyses in both the main and sensitivity analyses, we used the function kmeans.weight from the R package SWKM[74] in both cases (Supplementary Information Appendix C). Additionally, sensitivity analyses accounting for the distinct geographical extent of the data (global and continental) when delineating the biogeographical regions and calculating the biodiversity metrics also provided similar results (Supplementary Information Appendix D).

In the Supplementary Material, we provide R code for estimating the four biodiversity aspects of biota overlap, species richness, species occupancy and species endemism. This code also enables the clustering of grid cells based on these aspects to identify biogeographical sectors. Furthermore, the online tool Infomap Bioregions, for mapping biogeographical regions[54,61], now provides the values of these four biodiversity aspects (www.mapequation.org/bioregions2/).

## Neighbour analyses
To examine the spatial relationship among biogeographical sectors, we assessed whether these sectors exhibited a higher degree of

neighbourhood or adjacency to other sectors than expected by chance. With seven sectors, the expected chance probability of neighbouring a different sector was 1 in 6. To calculate the observed probability, we identified neighbouring grid cells belonging to different biogeographical sectors. Then, we evaluated whether the observed neighbouring probability surpassed the expected chance probability through binomial proportional tests. To account for variations in the number of neighbouring grid cells across different biogeographical sectors, we performed tests in both directions, assessing the neighbourhood between hypothetical sectors A and B (A → B and B → A). We deemed there was evidence of significant neighbourhood if either the A → B or B → A tests yielded $P < 0.05$. These analyses were carried out for each biogeographical region, depicting the frequency of instances where two sectors exhibited greater neighbouring than expected by chance. To ensure robust statistical analysis and avoid issues with expected values below five events in binomial proportional tests, we limited the analyses to combinations of biogeographical regions and sectors with over 30 neighbouring grid cells (five events or grid cells multiplied by the potential six neighbouring sectors). To perform the binomial proportion tests, we used the function prop.test in the R package stats[75]. The results are illustrated in Fig. 1c, where the circle sizes represent the relative proportion of statistically significant neighbouring events across sector pairs in all biogeographical regions and taxa. Supplementary Table 9 provides details on the total number of biogeographical regions, across the seven taxa, where the two compared biogeographical sectors were represented—that is, at least one grid cell from each of the two sectors. Supplementary Table 9 provides details on the proportion of those latter regions where neighbouring between pairs of sectors was higher than expected.

### Multinomial models
We examined whether the biogeographical sectors corresponded to geographical areas with distinct environmental conditions by using multinomial logit models with biogeographical sectors as the response variable. We assessed the existence of a correlation between the core-to-transition pattern with two widely used variables for explaining species diversity at the global scale: mean annual temperature and precipitation for the terrestrial biota and mean surface temperature and salinity for the marine biota. The data on mean annual temperature and precipitation were obtained from the Climatic Research Unit time series dataset v.4.06 (ref. [76]) and downscaled using WorldClim v.2.1 (ref. [77]), with the mean sea-surface temperature and sea-surface salinity data being obtained from the National Aeronautics and Space Administration's Ocean Color Web website (http://oceancolor.gsfc.nasa.gov/) and the National Oceanic and Atmospheric Administration's World Ocean Atlas 2009, and prepared by Sbrocco and Barber[78]. To align the raster data with our grid, we computed the mean values of corresponding pixels within each grid cell.

We performed multinomial models for each taxon and biogeographical region, comparing a model including the two respective explanatory variables against a null model including only an intercept. Evidence was considered statistically significant if the difference in the corrected Akaike information criterion value exceeded 10 (ref. [79]). We analysed biogeographical regions with two or more sectors. To mitigate issues with sample size in multinomial models, we only evaluated biogeographical sectors with more than 15 grid cells. To ensure the representativeness of the complete biogeographical region, we only evaluated those biogeographical regions where the sum of the biogeographical sectors with more than 15 grid cells represented at least 90% of the entire region. To assess the fit of the multinomial models, we used McFadden's pseudo-$R^2$. We performed multinomial models using the multinom function from the R package nnet[80], and we calculated McFadden's $R^2$ using the PseudoR2 function from the R package DescTools[81].

We also explored the potential correlation between present biodiversity and past climatic conditions (Supplementary Information Appendix A). We conducted sensitivity analyses to consider the spatial autocorrelation in the model residuals, obtaining similar results (Supplementary Information Appendix E).

### Taxonomic dissimilarity: nestedness versus turnover
We examined whether the variation in species composition among biogeographical sectors, measured by Sørensen pairwise dissimilarity, was more attributed to nestedness or to species turnover components[38]. In our case, nestedness refers to dissimilarity arising because the species composition in one biogeographical sector is a subset of the species found in another sector, whereas turnover captures dissimilarity due to species replacement[38]. For each bioregion and taxon, considering all present biogeographical sectors together, we computed the proportion of the taxonomic dissimilarity between sectors attributed to nestedness. This proportion was calculated as the ratio of the nestedness component to the total Sørensen dissimilarity. We calculated the distinct components using the beta.multi function from the R package betapart[82]. We calculated the nestedness for characteristic and non-characteristic species separately. Because the non-characteristic species can be affined to distinct biogeographical regions, we conducted independent analyses for the non-characteristic species affiliated with each biogeographical region.

### Local variation in species richness at the global scale
At our grid-cell resolution, variations in species richness were expected to depend on both the size of the regional species pool—resulting from the balance of speciation, extinction and biogeographical dispersal[45,46]—and the sorting of species from that pool[47,48]—in this study, associated with the core-to-transition organization and environmental filters. Previous studies has approximated the size of the regional species pool as the average species richness across locations within a biogeographical region[46]. Thus, the centred richness, calculated as the observed species richness in each grid cell minus the average richness in that biogeographical region, may be considered as representing species sorting within the region independently of the pool size. Therefore, to assess the relative importance of these factors, we modelled species richness across grid cells using three explanatory variables: (1) the total number of characteristic species per biogeographical region, obtained from the network analyses (a proxy for the size of the regional species pool); (2) the centred richness of characteristic species; and (3) the centred richness of non-characteristic species. These three variables provide an approximate decomposition of species richness. While these three variables together should explain a large proportion of the variance in species richness, this decomposition allowed us to explore the independent effect of each variable, and thus which regional effect was more relevant in explaining the global variance in local species richness. The size of the regional species pool would be more important in explaining global patterns in species richness if there were large differences in the size of the regional species pool among the biogeographic regions and the core-to-transition organization was, in general, driven by relatively few species. By contrast, the sorting of characteristic and non-characteristic species would be more important if there were small variations in the size of the regional species pools, and the core-to-transition organization was driven by relatively many species. These alternatives informed on whether global variations in species richness are more associated with the formation of the biogeographical regions and the species diversification and dispersals (size of regional species pool) or with the sorting of species due to environmental filters (core-to-transition organization).

To evaluate the relative importance of each factor, we fitted linear models of the grid-cell richness as a function of the three aforementioned variables. Then, we calculated the variance partitioning of

the three explanatory variables, focusing on the individual fraction explained by those—that is, the variance explained by a variable after removing the variance shared with the other explanatory variables. This individual fraction of variance of a given variable was measured as the difference between the $R^2$ of the saturated model and the $R^2$ of a model without the focal variable. To prevent inference biases stemming from larger biogeographical regions, we used weighted regression, where the cell values were weighted by the inverse size of the biogeographical region. The results from this sensitivity analysis were qualitatively similar using unweighted regression (Supplementary Fig. 35). We performed linear regressions using the lm function from the R package stats[75], and obtained the adjusted $R^2$ by applying the R function summary to the output of the linear regression models.

### Species richness in core area higher than expected

The core areas of each biogeographical region tend to comprise 30% of the biogeographical region while harbouring more species than the remaining 70%. This observation suggests a potential conservation value for these core areas[50]. To assess whether the species richness was higher than expected by chance, we conducted a randomization test for each combination of biogeographical region and taxon. We randomized the identity of the biogeographical sectors in the grid cells while maintaining the total number of grid cells per biogeographical sector and region. This process was repeated 100 times. We then evaluated whether the observed species richness in the core areas was higher than that in the randomized core.

### Reporting summary

Further information on research design is available in the Nature Portfolio Reporting Summary linked to this article.

## Data availability

The data supporting the findings of this study are publicly available from established repositories. The species distribution maps for amphibians, mammals, reptiles, rays and dragonflies were obtained from the International Union for Conservation of Nature (https://www.iucnredlist.org). The bird species distributions were sourced from Bird-Life International (https://www.birdlife.org) and the tree occurrence data from the United States Forest Inventory and Analysis Program (https://www.fia.fs.usda.gov). The climate data used in this study were obtained from the Climatic Research Unit time series dataset v.4.06 and WorldClim v.2.1 databases for the terrestrial taxa and from the National Aeronautical and Space Administration's Ocean Color Web website and the National Oceanic and Atmospheric Administration's World Ocean Atlas 2009 for the marine taxa. All raw distribution data are freely accessible for academic use upon request from the respective repositories or via their websites.

## Code availability

The R code used to calculate the biodiversity metrics, including the species richness, biota overlap, occupancy and endemicity, and to identify the biogeographical sectors through $k$-means clustering is provided as part of the Supplementary Information. The scripts are annotated and reproduce the core analytical procedures described in the manuscript. All scripts are intended for academic use and include documentation of input formats and parameter configurations.

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

## Acknowledgements

We thank J. Hortal, A. Benítez-López, V. Hermoso and C. Venditti's group for their comments on the text. We thank D. Edler for commenting on the text and incorporating the estimation of the four biodiversity aspects in the online tool Infomap Bioregions (www.mapequation.org/bioregions2/). We also thank J. Smiljanic for help with the stochastic block modelling analyses. We thank K. Stewart, P. Romero-Vidal and F. Hiraldo for discussions on this work. We also thank I. Afán and D. Aragonés for their help with grid-cell creation. Finally, we thank E. Gerber for his insights on residual estimation in multinomial models. R.B.-M. was supported by an Olle Engkvist Foundation grant (220-0175) and the Spanish Ministry of Economy and Competitiveness (BES-2013-065753). M. Rosvall was supported by the Swedish Research Council under grant 2023-03705. J.A. was supported by the Basque Government's Postdoctoral Programme for the Improvement of Doctoral Research Staff (POS_2024_1_0026) and by Basque Government funding for the FisioKlima-AgroSosT research group (IT1682–22). E.R. was funded by the Spanish Ministry of Economy and Competitiveness (CGL2012-35931/BOS). M. Rueda was funded by European Union project 2020/125-US/JUNTA/FEDER EU (Programa Operativo FEDER/Junta de Andalucía 2014–2020). This research was also funded by the Spanish Ministry of Science and Innovation (MICINN) through the European Regional Development Fund (SUMHAL, LIFEWATCH-2019-09-CSIC-13, POPE 2014–2020) and the Ministry of Universities through the University of Seville, as part of the call for the Requalification of the Spanish University System 2021–2023, funded by the European Union's NextGenerationEU.

## Author contributions

R.B.-M. and J.C. conceived the study. R.B.-M., M.G.-S., J.C., M. Rueda and J.A. curated the data. R.B.-M., J.C. and M. Rosvall developed the biodiversity metrics. R.B.-M. and J.C. designed the statistical framework. R.B.-M. performed the data analysis and produced all the figures and visualizations, with contributions from J.C. and M.G.-S. R.B.-M., J.C. and M.G.-S. developed the manuscript outline. R.B.-M. wrote the first draft. All authors provided critical revisions and approved the final version of the manuscript.

## Funding

## Competing interests

The authors declare no competing interests.

## Additional information

**Extended data** is available for this paper at https://doi.org/10.1038/s41559-025-02724-5.

**Correspondence and requests for materials** should be addressed to R. Bernardo-Madrid.

[1]Integrated Science Lab, Department of Physics, Umeå University, Umeå, Sweden. [2]Departmento de Biología Vegetal y Ecología, Universidad de Sevilla, Seville, Spain. [3]Ecology and Evolutionary Biology, School of Biological Sciences, University of Reading, Reading, UK. [4]Department of Conservation Biology, Doñana Biological Station, Consejo Superior de Investigaciones Científicas (CSIC), Seville, Spain. [5]Department of Physical, Chemical and Natural Systems, Universidad Pablo de Olavide, Seville, Spain. [6]Universidad de Alcalá, Department of Life Sciences, Forest Ecology and Restoration Group (FORECO), Alcalá de Henares, Spain. [7]Department of Physical Geography and Ecosystem Science, Lund University, Lund, Sweden. [8]Departmento de Biología y Geología, Física y Química Inorgánica, Escuela Superior de Ciencias Experimentales y Tecnología (ESCET), Universidad Rey Juan Carlos, Madrid, Spain. [9]Global Change Research Institute, Rey Juan Carlos University, Madrid, Spain. ✉e-mail: ruben.bernardo.madrid@umu.se

**Step 1**
Spatial projection

**Step 2**
Bipartite network

**Step 3**
Network analyses

**Step 4**
Bioregion and regional biota from modules
(Extended Data Figs. 2-8)

Characteristic species

Non-characteristic species

Characteristic species

Non-characteristic species

**Step 5**

Four biodiversity metrics based on modules and links from steps 3-4 (Supplementary Figs. S1-7)

| Relative species richness of characteristic species | Biota overlap | Mean relative occupancy of characteristic species | Mean endemicity of characteristic species |
|---|---|---|---|
| cell | cell | species | species |
| z-score of N intra-module links | prop. links outside the module | z-score of N intra-module links | prop. links within the module |

$$\frac{(I_c - \bar{I}_{m_c})}{\sigma I_{m_c}}$$

$$\frac{O_c}{L_c}$$

$$Mdn\left(\frac{I_s}{L_s}\right)$$

$$Mdn\left(\frac{I_s - \bar{I}_{m_c}}{\sigma I_{m_s}}\right)$$

Example Link types

cells

$I_c = \sum I$
$O_c = \sum O$
$L_c = O_c + I_c$

species

$I_s = \sum I$
$O_s = \sum O$
$L_s = O_s + I_s$

**Numerical calculation based on step 3**

**Biota overlap (grid cells)**
A1 = 0/1 = 0
A2 = 0/3 = 0
B1 = 1/3 = 0.33
B2 = 1/3 = 0.33
C1 = 0/2 = 0
C2 = 0/1 = 0

**Endemicity (species)**
Lion  = 2/2 = 1
Giraffe = 3/4 = 0.75
Rhino  = 1/1 = 1
Moose = 3/3 = 1
Bear  = 2/3 = 0.66

**Median endemicity (grid cells)**
A1 = (0.75)/1  = 0.75
A2 = (1+0.75+1)/3  = 1
B1 = (1+0.75)/2  = 0.875
B2 = (1+0.66)/2  = 0.83
C1 = (1+0.66)/2  = 0.83
C2 = (1)/1  = 1

**Relative species richness (grid cells)**
A1 = (1-2) / 1 = -1
A2 = (3-2) / 1 = 1
B1 = (2-2) / 1 = 0
B2 = (2-1.66) / 0.57 = 0.60
C1 = (2-1.66) / 0.57 = 0.60
C2 = (1-1.66) / 0.57 = -1.58

Mean spp richness Brown = 2
sd spp richness Brown = 1

Mean spp richness Green = 1.66
sd spp richness Green= 0.57

**Relative occupancy (species)**
Lion  = (2-2) / 1  = 0
Giraffe = (3-2) / 1  = 1
Rhino  = (1-2) / 1  = -1
Moose = (3-2.5) / 0.7 = 0.71
Bear  = (2-2.5) / 0.7 = -0.71

Mean occupancy Brown = 2
sd occupancy Brown = 1

Mean occupancy Green = 2.5
sd occupancy Green= 0.7

**Median relative occupancy (grid cells)**
A1 = (1) / 1  = 1
A2 = (0 + 1 + -1) / 3  = 0
B1 = (0 + 1) / 2  = 0.5
B2 = (0.71 + -0.71) / 2 = 0
C1 = (0.71 + -0.71) / 2 = 0
C2 = (0.71 ) / 1  = 0.71

**Step 6**
Identify biogeographical sectors: Cluster grid cells based on scaled biodiversity metrics (Extended Data Figs. S2-8)

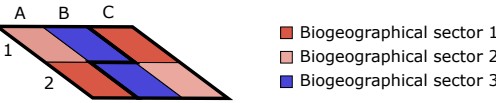

■ Biogeographical sector 1
■ Biogeographical sector 2
■ Biogeographical sector 3

**Extended Data Fig. 1 | See next page for caption.**

**Extended Data Fig. 1 | Workflow to obtain biogeographical regions, characteristic and non-characteristic species and biogeographical sectors.** (Step 1) Hypothetical species distribution in grid cells. We called grid cells using chess terminology (columns letters and rows numbers). (Step 2) Representation of species distribution in a network format. Species and grid cells are different types of nodes where links represent the occurrence of a species in a grid cell. (Step 3) Output of the community detection algorithm (also called network clustering algorithms) as a set of modules, represented with distinct colours, containing the nodes assigned. In the case of Infomap, this clustering is based on Map Equation a flow-based and information-theoretic method[23,24]. See Supplementary Information of Bernardo-Madrid et al.[19] for a non-specialized description. (Step 4) Visualization of biogeographical regions, characteristic species, and non-characteristic species. (Step 5) Calculation of the four-biodiversity metrics based on network cartography. (Step 6) Clustering of the grid cells with a k-means using the four-biodiversity metrics scaled as features.

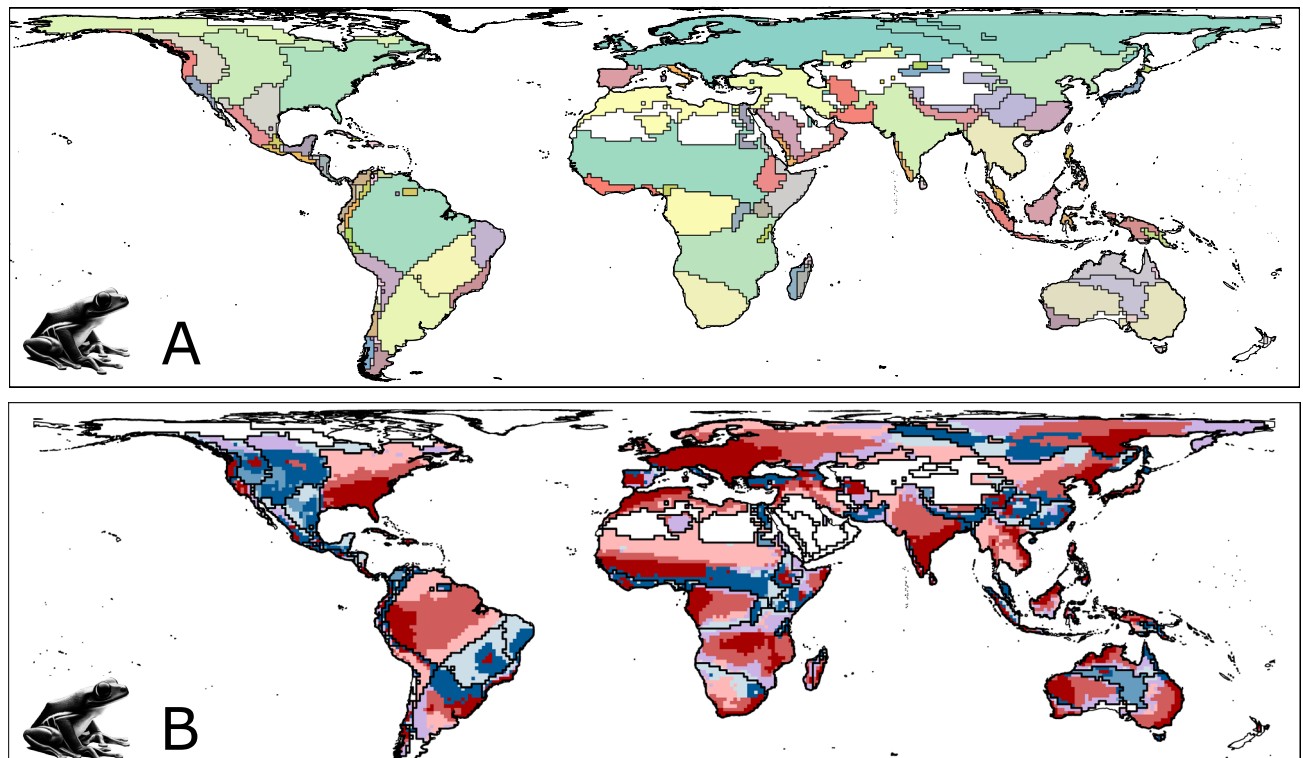

**Extended Data Fig. 2 | Biogeographical regions and sectors of amphibians.**
(**a**) Amphibian's bioregions. White areas indicate geographical regions where this taxon is absent, or grid cells overlap with ≤ 50% of the Earth's surface. (**b**) General biogeographical sectors within amphibian bioregions. Solid black lines represent the boundaries of the biogeographical regions, while colours denote different biogeographical sectors. Warmer colours indicate geographical areas with relatively low overlap of biotas, reflecting distinct biogeographical affinities (i.e. overlap of distinct characteristic species pools). Cooler colours represent areas with higher overlap. Darker and lighter shades within the warm and cold colour ranges, respectively, indicate high and low richness of characteristic species. See Fig. 1 for relationship between the biogeographical sectors and the four-biodiversity metrics. Some biogeographical regions, mostly depicted by blue colours, may represent widely recognized transitional zones. Nonetheless, these biogeographical regions can also correspond to lower biogeographical hierarchical levels, such as subregions or domains[58]. The identification of these highly transitional biogeographical regions opens possibilities for clustering them into higher biogeographical hierarchical scales, such as the Western subregion of the Neartic or the Chacoan subregion of the Neotropical[58]. However, to ensure objectivity in delineating bioregions and avoid subjective criteria for thresholds[9,56], all bioregions with statistical support were studied independently of their potential biogeographical hierarchical level[58].

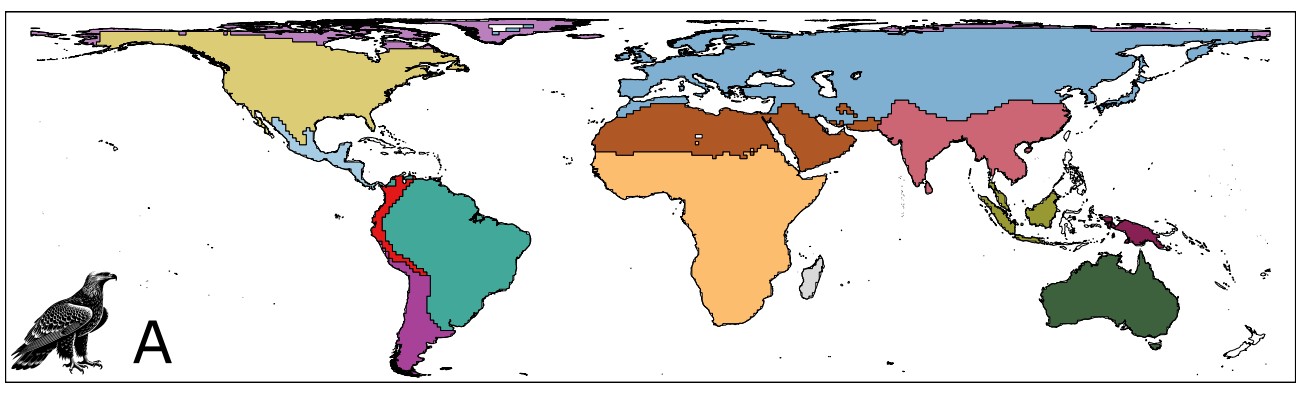

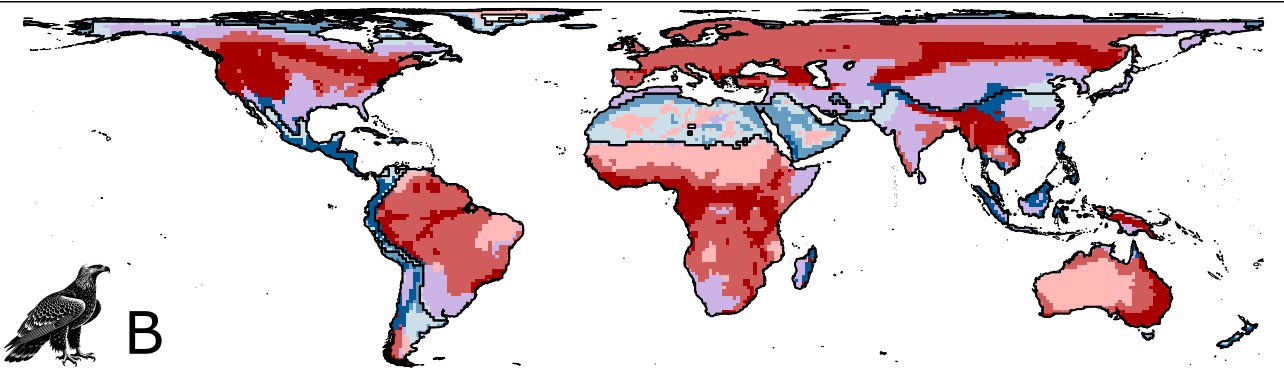

**Extended Data Fig. 3 | Biogeographical regions and sectors of birds.** (**a**) Bioregions of birds. (**b**) Biogeographical sectors within biogeographical regions of birds. See details on caption of Fig. 1 and Extended Data Fig. 2.

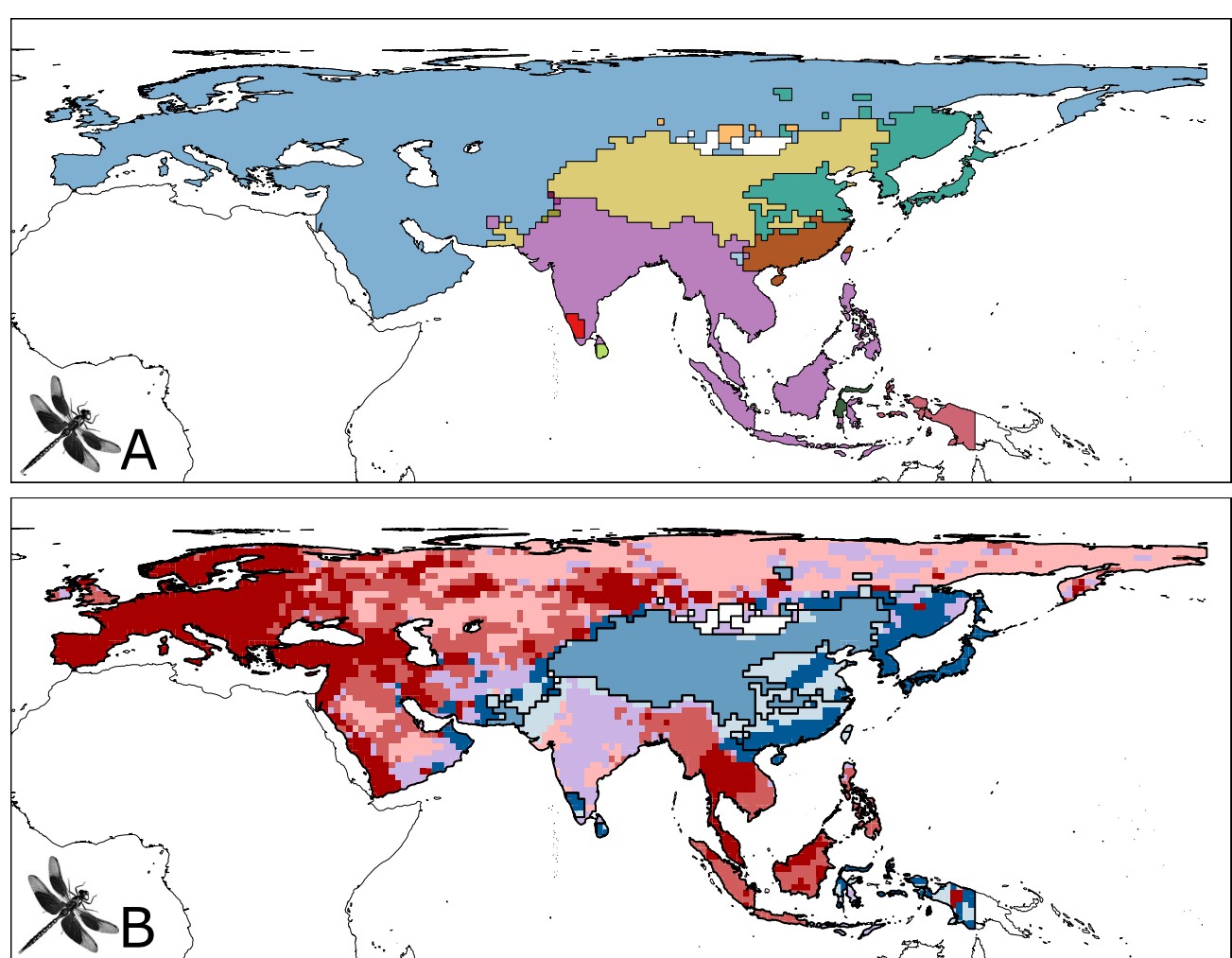

**Extended Data Fig. 4 | Biogeographical regions and sectors of dragonflies. (a)** Bioregions of dragonflies. (**b**) Biogeographical sectors within biogeographical regions of dragonflies. See details on caption of Fig. 1 and Extended Data Fig. 2.

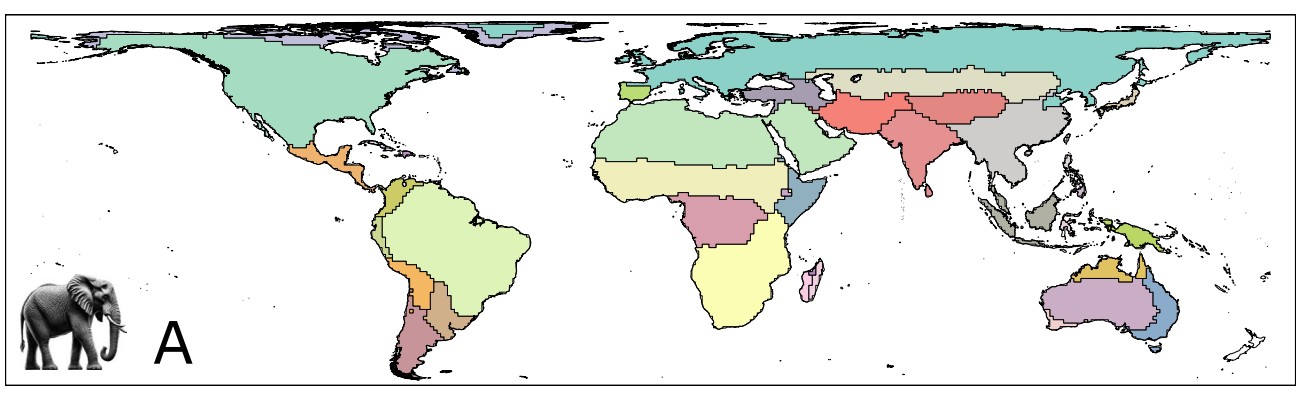

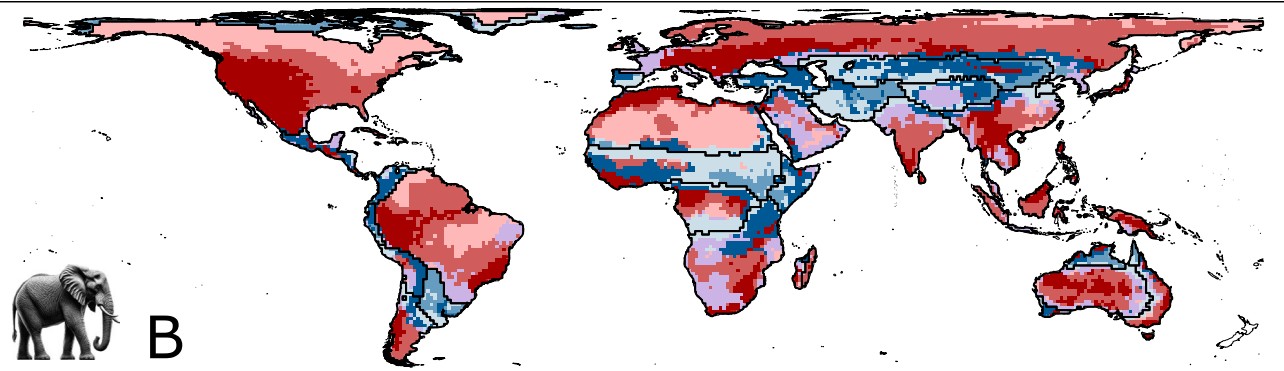

**Extended Data Fig. 5 | Biogeographical regions and sectors of mammals. (a)** Bioregions of mammals. **(b)** Biogeographical sectors within biogeographical regions of mammals. See details on caption of Fig. 1 and Extended Data Fig. 2.

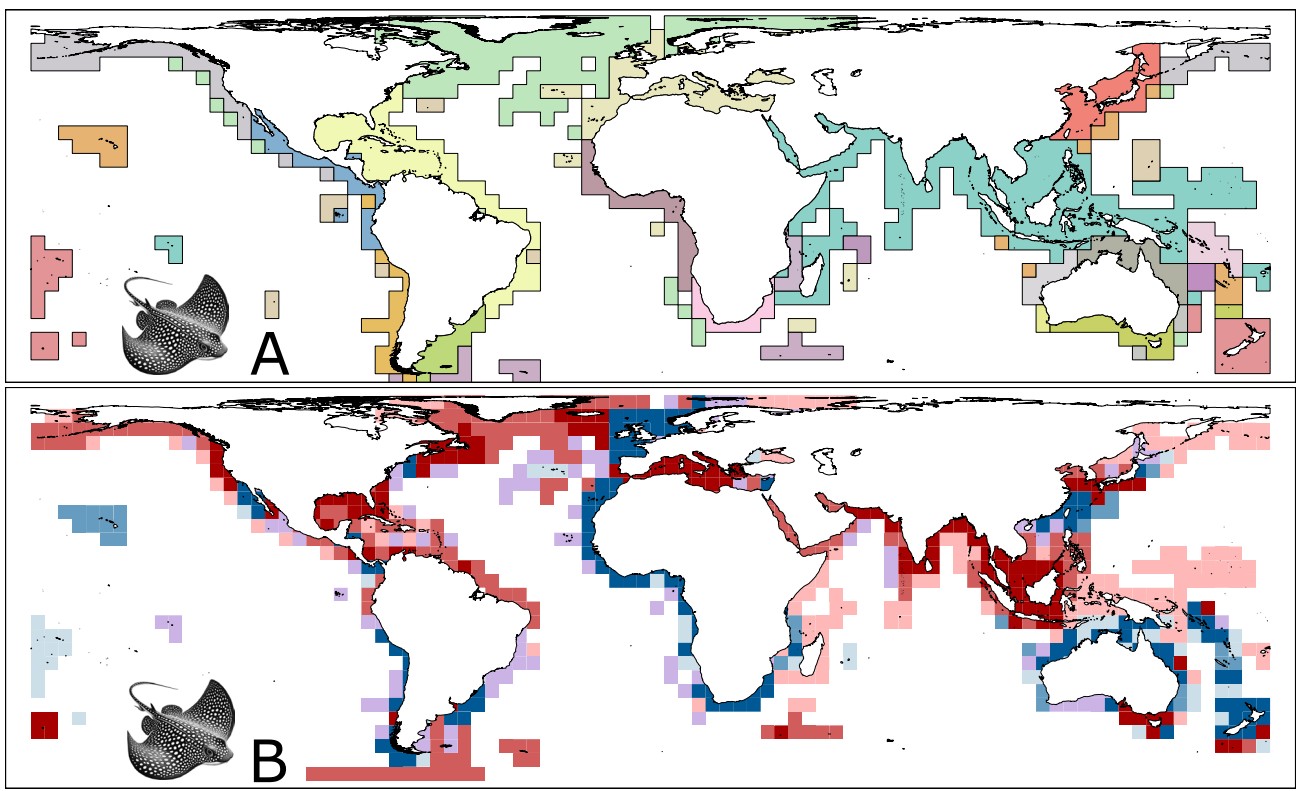

**Extended Data Fig. 6 | Biogeographical regions and sectors of rays.** (**a**) Bioregions of rays. (**b**) Biogeographical sectors within biogeographical regions of rays. See details on caption of Fig. 1 and Extended Data Fig. 2.

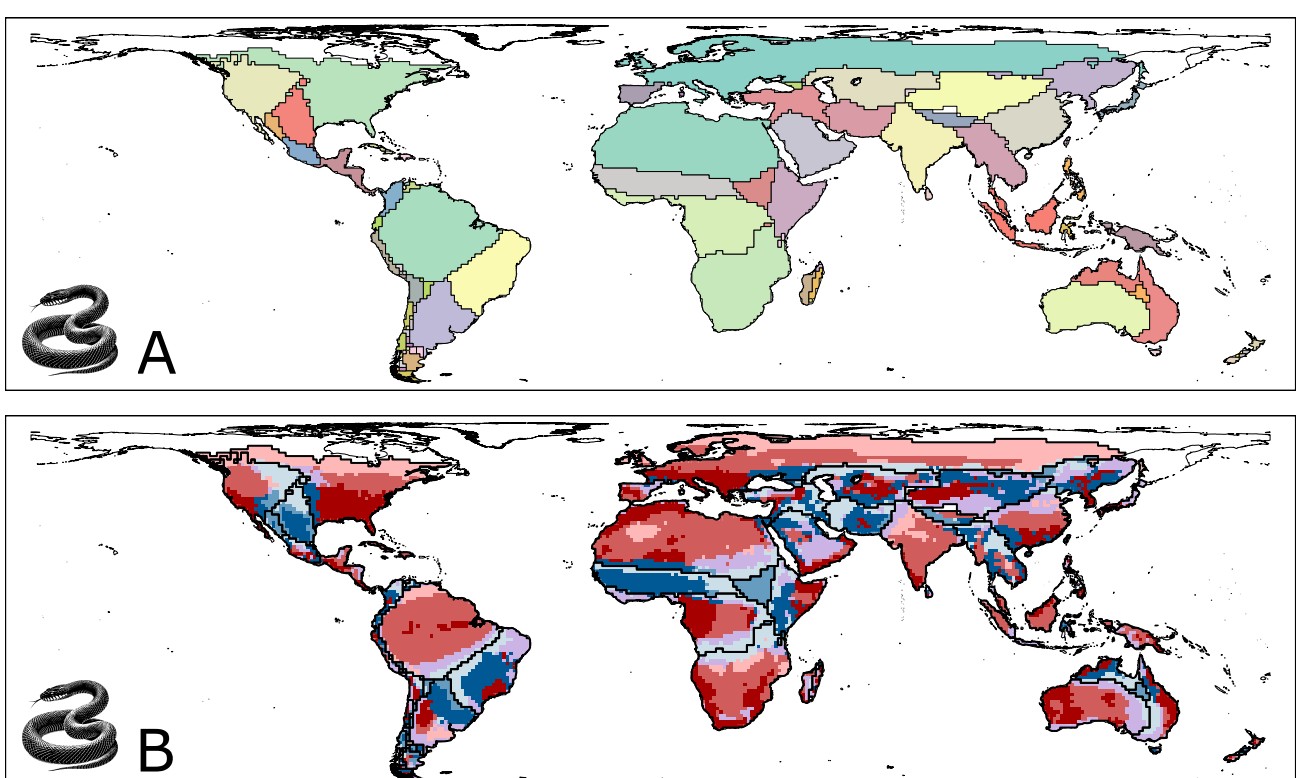

**Extended Data Fig. 7 | Biogeographical regions and sectors of reptiles.** (**a**) Bioregions of reptiles. (**b**) Biogeographical sectors within biogeographical regions of reptiles. See details on caption of Fig. 1 and Extended Data Fig. 2.

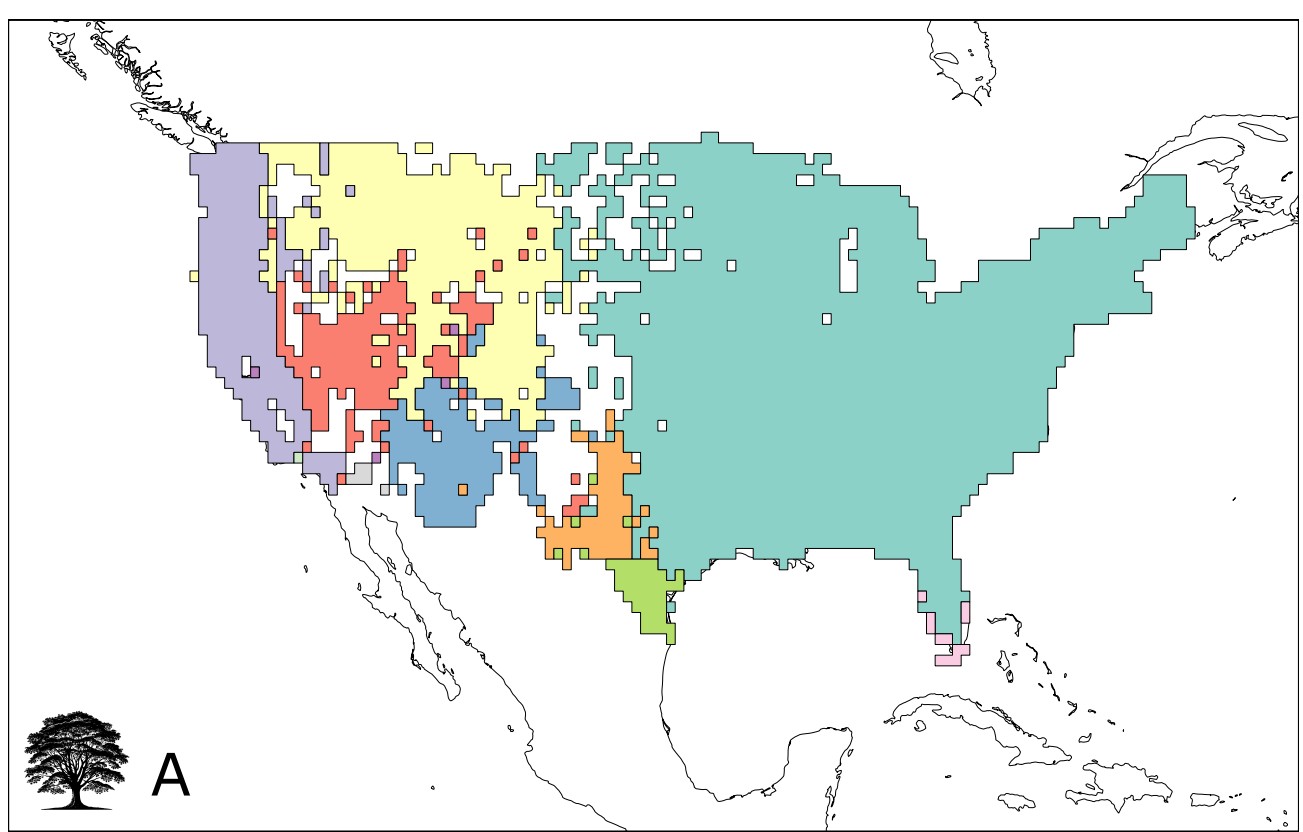

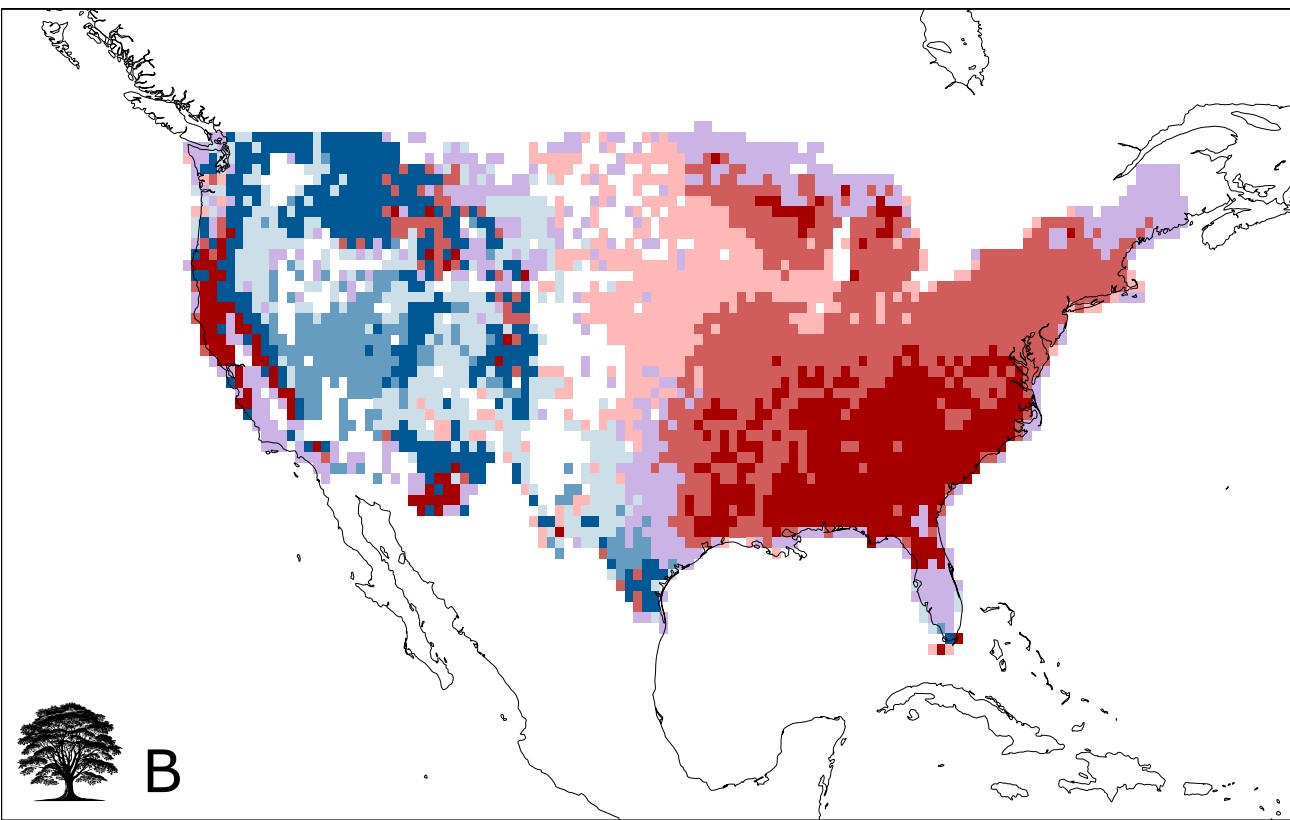

**Extended Data Fig. 8 | Biogeographical regions and sectors of trees.** (**a**) Bioregions of trees. (**b**) Biogeographical sectors within biogeographical regions of trees. See details on caption of Fig. 1 and Supplementary Fig. 2.

ruben.bernardo.madrid@umu.se

# Reporting Summary

## Statistics

For all statistical analyses, confirm that the following items are present in the figure legend, table legend, main text, or Methods section.

| n/a | Confirmed | |
|---|---|---|
| ☐ | ☒ | The exact sample size (*n*) for each experimental group/condition, given as a discrete number and unit of measurement |
| ☒ | ☐ | A statement on whether measurements were taken from distinct samples or whether the same sample was measured repeatedly |
| ☐ | ☒ | The statistical test(s) used AND whether they are one- or two-sided<br>*Only common tests should be described solely by name; describe more complex techniques in the Methods section.* |
| ☐ | ☒ | A description of all covariates tested |
| ☒ | ☐ | A description of any assumptions or corrections, such as tests of normality and adjustment for multiple comparisons |
| ☐ | ☒ | A full description of the statistical parameters including central tendency (e.g. means) or other basic estimates (e.g. regression coefficient) AND variation (e.g. standard deviation) or associated estimates of uncertainty (e.g. confidence intervals) |
| ☒ | ☐ | For null hypothesis testing, the test statistic (e.g. *F*, *t*, *r*) with confidence intervals, effect sizes, degrees of freedom and *P* value noted<br>*Give P values as exact values whenever suitable.* |
| ☒ | ☐ | For Bayesian analysis, information on the choice of priors and Markov chain Monte Carlo settings |
| ☒ | ☐ | For hierarchical and complex designs, identification of the appropriate level for tests and full reporting of outcomes |
| ☐ | ☒ | Estimates of effect sizes (e.g. Cohen's *d*, Pearson's *r*), indicating how they were calculated |

*Our web collection on statistics for biologists contains articles on many of the points above.*

## Software and code

Policy information about availability of computer code

| Data collection | Data used is provided free of charge by IUCN, BirdLife, and Forest Inventory of United States (FIA) |
|---|---|
| Data analysis | All analyses were run in R software (Version 4.3.0). R code and objects are provided in the SI<br>R package SWKM version 0.09<br>R package aricode function AMI version 1.0.3<br>R package nnet Version 7.3-18<br>R package nnet Version 0.99.51<br>R package betapart Version 1.6<br><br>Infomap https://www.mapequation.org/infomap/  Version 2.6.1 |

For manuscripts utilizing custom algorithms or software that are central to the research but not yet described in published literature, software must be made available to editors and reviewers. We strongly encourage code deposition in a community repository (e.g. GitHub). See the Nature Portfolio guidelines for submitting code & software for further information.

## Data

The data supporting the findings of this study are publicly available from established repositories. Species distribution maps for amphibians, mammals, reptiles, rays, and dragonflies were obtained from the International Union for Conservation of Nature (IUCN; https://www.iucnredlist.org). Bird species distributions were sourced from BirdLife International (https://www.birdlife.org), and tree occurrence data from the United States Forest Inventory and Analysis Program (FIA; https://www.fia.fs.usda.gov). Climate data used in this study were obtained from the CRU-TS 4.06 and WorldClim 2.1 databases for terrestrial taxa, and from NASA's Ocean Color and NOAA's World Ocean Atlas 2009 for marine taxa. All raw distribution data are freely accessible for academic use upon request from the respective repositories or via their websites.

## Research involving human participants, their data, or biological material

| | |
|---|---|
| Reporting on sex and gender | not apply |
| Reporting on race, ethnicity, or other socially relevant groupings | not apply |
| Population characteristics | not apply |
| Recruitment | not apply |
| Ethics oversight | not apply |

Note that full information on the approval of the study protocol must also be provided in the manuscript.

# Field-specific reporting

Please select the one below that is the best fit for your research. If you are not sure, read the appropriate sections before making your selection.

☐ Life sciences    ☐ Behavioural & social sciences    ☒ Ecological, evolutionary & environmental sciences

For a reference copy of the document with all sections, see [nature.com/documents/nr-reporting-summary-flat.pdf](nature.com/documents/nr-reporting-summary-flat.pdf)

# Ecological, evolutionary & environmental sciences study design

All studies must disclose on these points even when the disclosure is negative.

| | |
|---|---|
| Study description | Analyses on biodiversity patterns based on distribution range maps of amphibians, birds, dragonflies, mammals, rays, and reptiles, as well as forest inventory of United States |
| Research sample | 30,049 marine and terrestrial vertebrates, invertebrates and plant species |
| Sampling strategy | Not apply |
| Data collection | Downloaded from IUCN, BirdLife and FIA websites |
| Timing and spatial scale | Data downloaded in 2021. The extend of the data is planetary |
| Data exclusions | not apply |
| Reproducibility | Freely available data and R code provided |
| Randomization | not apply |
| Blinding | not apply |

Did the study involve field work?  ☐ Yes  ☒ No

# Reporting for specific materials, systems and methods

We require information from authors about some types of materials, experimental systems and methods used in many studies. Here, indicate whether each material, system or method listed is relevant to your study. If you are not sure if a list item applies to your research, read the appropriate section before selecting a response.

## Materials & experimental systems

| n/a | Involved in the study |
|-----|------------------------|
| ☒ ☐ | Antibodies |
| ☒ ☐ | Eukaryotic cell lines |
| ☒ ☐ | Palaeontology and archaeology |
| ☒ ☐ | Animals and other organisms |
| ☒ ☐ | Clinical data |
| ☒ ☐ | Dual use research of concern |
| ☒ ☐ | Plants |

## Methods

| n/a | Involved in the study |
|-----|------------------------|
| ☒ ☐ | ChIP-seq |
| ☒ ☐ | Flow cytometry |
| ☒ ☐ | MRI-based neuroimaging |

## Plants

Seed stocks — not apply

Novel plant genotypes — not apply

Authentication — not apply

