## [Peer Review File · Nature Ecology & Evolution]

A general rule on the organization of biodiversity on Earth's biogeographical regions

Corresponding Author: Dr Rubén Bernardo-Madrid

Version 0:

Decision Letter:

14th June 2024

Dear Dr Bernardo-Madrid,

First of all, my sincere apologies for the delay with your manuscript. Unfortunately, one of our three reviewers has failed to submit a report and we have been unable to contact them. We therefore need to proceed with 2 reviewers at this stage, but because we still need a reviewer with methodological expertise, we will need to recruit a new reviewer at the next round. From the two reports below, you will see that the current reviewers have raised a number of concerns which will need to be addressed before we can offer publication in Nature Ecology & Evolution.

We therefore invite you to revise your manuscript taking into account all reviewer comments. Please highlight all changes in the manuscript text file.

* If you have not done so already please begin to revise your manuscript so that it conforms to our Article format instructions at <http://www.nature.com/natecolevol/info/final-submission>. Refer also to any guidelines provided in this letter.

Link Redacted

Nature Ecology & Evolution is committed to improving transparency in authorship. As part of our efforts in this direction, we are now requesting that all authors identified as 'corresponding author' on published papers create and link their Open Researcher and Contributor Identifier (ORCID) with their account on the Manuscript Tracking System (MTS), prior to acceptance. ORCID helps the scientific community achieve unambiguous attribution of all scholarly contributions. You can create and link your ORCID from the home page of the MTS by clicking on 'Modify my Springer Nature account'. For more

information please visit please visit www.springernature.com/orcid.

[redacted]

Reviewers' comments:

Reviewer #1 (Remarks to the Author):

This manuscript combines biogeographical analyses of multiple taxa (plants, vertebrates and dragonflies) to identify generalities in the biogeographical organization of species distribution on Earth.

The key message of the manuscript is that the authors found a rather general pattern that can be defined as a transition from "core" areas to transition areas.

This idea is somehow implicit to many classical biogeographical analyses (see Kreft, H., and W. Jetz. 2013. Comment on "An Update of Wallace's Zoogeographic Regions of the World". *Science* 341:343; see also previous work such as Müller, P. 1974. *Aspects of Biogeography*. Dr. W. Junk b.v. Publishers, The Hague. Müller, P. (1986). *Biogeography*. Harper & Row). What is novel, is the attempt to retrieve these pattern from quantitative biogeographical analyses.

I appreciated several aspects of this manuscript, including the multi-taxa approach and the large number of analyses performed to support the conclusions.

Nevertheless, the manuscript is not entirely clear. There are several points where it is not clear what the authors are actually doing, and why. In some cases this made it difficult assessing the suitability of the manuscript (i.e. in some cases it was not clear if the authors used appropriate methods / followed a strategy that is appropriate to achieve its aims).

In my view, several clarifications are pivotal to make the manuscript fully understandable

Specific comments

In some cases the manuscript exaggerates its generality. For instance, the title (A general rule on the organization of biodiversity on Earth) is too vague. It seems that you have found a general rule for all biological sciences. It must be clear that the manuscript focuses on biogeography and on species distributions / biogeographical regions

The first paragraph of the introduction is very clear

L 35: this sentence is a bit awkward, please rephrase

L 36: It is increasingly recognized that no bioregion is "virtually devoid" of life forms. Even Antarctica is increasingly recognized as an environment hosting a significant biodiversity

L 43 (and other parts of the manuscript): here it is quite unclear. Are you focusing on biogeographical or on ecological organization?

Results:

The identification of the 7 "roles" is the key result of the manuscript, but note that the generality is at least partially related to the fact that the authors force the 7 roles for all the taxa (in fact the optimal number of roles can be quite different across taxa, see Supplementary Table 9)

the interpretation of the 7 "roles" should be explained in the text, or in a clearer version of Fig 1 (or perhaps both...). What do you mean eg by role 1? (eg role 1 = high endemism, low biota overlap, high relative species richness, low average occupancy...). I understand that writing everything in the text can make it the results length, but the authors should give to readers a key for a faster interpretation of Fig 1A (at least to me, understanding these roles was not very easy).

Fig 1 is central to the understanding of the manuscript

I do not think the scale is appropriate. This color scheme gives the impression of a linear gradient, but we do not have a continuous gradient. I suggest using a discrete color scheme that gives more the idea of clusters. In fact in the text (eg L 90-99) you nicely represent these roles using rather clear terms (eg "hotspots" vs "transitional areas"). It would be nice associating these terms to the 7 roles. A discrete color scheme allows a better identification of the roles on the map (this would be important in both Fig 1 and Fig 2). Consider approaches such as the one used in the supplementary figures S1-S2 of Holt et al (2013 *Science*).

NMDS-like approaches might also be used to clarify the interpretation of "roles" along the core-transition organization

L 104: see also Kreft, H., and W. Jetz. 2013. Comment on "An Update of Wallace's Zoogeographic Regions of the World". *Science* 341:343 and references therein

L 165: Please be more explicit about your predictions

L 173-174: Is 90% more than what would be expected by chance? This can be tested by creating "null" core areas having

the same average size of the true core areas and calculating the % hosted species (and repeating this a number of times).

Paragraph "influence across spatial scales"

In general, this is the less clear paragraph of the manuscript. The rationale of the analyses is not very clear to me and should be better explained. A key result is that the (standardized) number of characteristic species is a good predictor of the total species richness of a cell. Is this surprising? If I correctly interpreted this analysis, this seems to be a quite obvious expectation.

Furthermore, from the text (and the methods) I had the impression that the 3 variables were included jointly in the same analysis as independent, but I guess this is not the case, as the summed % explained variance (pool size + characteristic + non-characteristic species) sometimes reach values close to 100%

L 197: the variance of what? Please better explain the dependent and independent variables

L 211-212: note that you did not test if this holds across all the bioregions – you calculated average values across all the bioregions, but it is possible that this holds for some bioregions and not for other bioregions

L 212-213: this sentence needs references

Conclusive paragraph:

L 233-235: this is the reason why biodiversity hotspots were originally defined (Myers et al 2000. Biodiversity hotspots for conservation priorities. Nature 403:853-858)

L 356-357: this explanation of non-characteristic species is too long and makes it the sentence unclear. I suggest explaining well the definition of non-characteristic species above (around L 346-348), and here just using the term "non characteristic species"

Formulas used to calculate the 4 metrics should be explicit in Fig S1.

For instance, in Fig S1-step 5 - relative species richness, it should be clear that it is calculated as $(\text{cell species richness} - \text{mean}(\text{SR of the bioregion})) / (\text{SD}(\text{SR of the bioregion}))$

And so on.

Elsewhere it is very tricky to understand where do 1, 2, 0.57... come from

Fig S1 it is quite important to understand the meaning of the 4 metrics (particularly of occupancy). Consider improving this figure, and moving it to the main text

L 402; this should be Supplementary Tables 1-8. Also note that the absolute value of r should be the relevant value here

L 409: tiny biogeographical regions...

L 413-416: Please explain why you used this approach, and not alternative approaches to the selection of the number of clusters. Consider showing the plots of goodness-of-fit and the detected thresholds in a supplementary figure

L 421: to be honest, the Calinski-Harabasz index produced rather different results (eg Amphibians: piecewise regression 8, Calinski-Harabasz index 2; Rays: 8 vs. 3)

Also in this case, showing the plots may help to evaluate the robustness of selected thresholds

L 422-423: this is really unclear to me. How did you cluster all cells together for the global analysis? The spatial extent (and even the grain) is very different across taxa

L 431: please add references for the AMI. I acknowledge that I'm not an expert of this approach

L 511: here do you mean dissimilarity between cells within bioregion? Or between "roles" within bioregions?

Reviewer #3 (Remarks to the Author):

I find that the manuscript represents a very interesting work. I will not comment on the methods, but on a few issues that I think may help improve the manuscript.

"Non-characteristic species" (e.g., lines 67, 72, etc.): I am not sure if this means non-endemic species? This should be clarified.

"Roles" (e.g., lines 78, etc.): this word seems a little awkward to me. I suggest the authors to provide a better one.

I find that the "Extended Data Figures 2-8" represent a great illustration of the results, and I don't think it is appropriate to have them relegated to an appendix. I suggest the authors to incorporate them to the main text.

Juan J. Morrone

*****END*****

Version 1:

Decision Letter:

Dear Dr Bernardo-Madrid,

Thank you for your patience while your revised manuscript "A general rule on the organization of biodiversity on Earth's biogeographical regions" was under review. We have now received comments from two reviewers. You will see from their comments copied below that while they find your work of considerable potential interest and improved in revision, but there are some important points to address. In light of these comments, we will need to see a revised version to make a decision on publication.

We hope you will find the reviewers' comments useful as you decide how to proceed. Please do not hesitate to contact us if there are specific requests from the reviewers that you believe are technically impossible or unlikely to yield a meaningful outcome. However, we expect the revision to feature additional analyses with state-of-the-art methods to address Reviewer 4's main concerns, as well as Reviewer 1's query on how spatial autocorrelation was accounted for.

* Highlight all changes in the manuscript text file and provide a version in Microsoft Word format, with line numbers.

* Include a "Response to reviewers" document detailing, point-by-point, how you addressed each referee comment. If no action was taken to address a point, you must provide a compelling argument. This response will be sent back to the referees along with the revised manuscript.

* If you have not done so already we suggest that you begin to revise your manuscript so that it conforms to our Article format instructions at <http://www.nature.com/natecolevol/info/final-submission>. Refer also to any guidelines provided in this letter.

Link Redacted

If you wish to submit a suitably revised manuscript we would hope to receive it within 3 months. If you cannot send it within this time, please let us know. We will be happy to consider your revision so long as nothing similar has been accepted for publication at Nature Ecology & Evolution or published elsewhere.

Nature Ecology & Evolution is committed to improving transparency in authorship. As part of our efforts in this direction, we are now requesting that all authors identified as 'corresponding author' on published papers create and link their Open Researcher and Contributor Identifier (ORCID) with their account on the Manuscript Tracking System (MTS), prior to acceptance. This applies to primary research papers only. ORCID helps the scientific community achieve unambiguous attribution of all scholarly contributions. You can create and link your ORCID from the home page of the MTS by clicking on 'Modify my Springer Nature account'. For more information please visit please visit www.springernature.com/orcid.

Thank you for the opportunity to review your work.

[redacted]

Reviewers' comments:

Reviewer #1 (Remarks to the Author):

The authors performed a considerable work to improve their manuscript, and I appreciated the efforts to perform new analyses (eg the sensitivity analysis). The introductive and conclusive part, as well as result interpretation, are generally clear.

Nevertheless there are a few points where some clarifications are needed

For me, the most confusing aspect of the manuscript is how the results from different taxa (different areas, different resolution...) were merged using clustering and AMI. This is explained in the methods (please note that I'm not an expert of these approaches; which values are used here?). However, I think that the main text (eg par "general organization of biodiversity") requires a short description of the fact that maps of multiple taxa were summarized to produce results in Fig 1 A-C

L 101-103: this sentence is not very clear to me. What do you mean by "one taxon"? do you refer to the comparison between the analyses performed on different "taxa" (eg birds vs amphibians) on the same cells? Or something different?

L 105: Which result? I'm missing a sentence assessing the similarity between taxa

Fig 1C reports an interesting conclusion that can be more explicitly reported in the text. What is the proportion of significant tests?

Multinomial models suggest that environmental features (eg precipitation and temperature) explain a good proportion of the assignation to biogeographical sectors within bioregion.

In principle, this can be also caused by spatial autocorrelation (both climate and species distribution are affected by strong spatial autocorrelation). Are these results consistent if you take into account spatial autocorrelation into your models? The same issue applies to the use of past climate.

Spatial autocorrelation may be relevant also for the linear models described at L 654 (part of variation is probably just explained by the spatial component)

Minor comments:

Fig 2: the inset is too small.

Reviewer #4 (Remarks to the Author):

The authors make an analysis of the organization of biodiversity in biogeographical regions using mostly a networks approach. I agree with one of the reviewers who says that while the ideas presented in the manuscript are not necessarily new, the manuscript offers quantitative evidence for those ideas.

Nonetheless, I have been asked to make a technical assessment on the manuscript, so I will make comments on those technical aspects of the study that I think deserve special attention. That being said, I know that this is at least a second round of review, but I do think that these few points should be addressed before making a final decision on the suitability for publication of this manuscript.

Data: The authors pool together different datasets corresponding to different types of fauna and even plants. However, not all of the datasets cover the same region worldwide. For instance, the data on trees is only available for North America. My question here is whether the results are affected by the fact that not all data is equally distributed geographically and how that affects the biogeographical regions the authors identify and the metrics the authors define.

Obtention of biogeographical regions: To determine 'biogeographical' regions in the bipartite network defined by geographical regions and species, the authors use Infomap, a method that aims at finding groups that contain species and geographical regions together. However, the authors do not justify this choice. In the networks literature, there are many papers that talk about detecting 'communities in bipartite networks' by finding groups of species and groups of geographical areas 'separately' (see at least papers from PRE 2007 – Barber and Guimera et al). I am not sure why the authors did not adopt this methodology instead, since it seems more appropriate for the type of network the authors deal with than the approach used in the manuscript. Indeed, there are more modern approaches that do not assume that groups of species (or communities) need to comprise species that 'are more likely to be connected to the same groups of geographical areas'. Indeed other connectivity patterns of the bipartite network might explain the patterns we observe, so it seems not as compelling to find these biogeographical regions when using a method that is looking for precisely this type of signal (i.e densely connected groups of geographical areas and species) that a method that is not (see for instance Gerlach et al Science Adv. 2018 or Cobo-López et al PNAS Nexus 2022). I think it is important for the authors to check the consistency of their results. Right now, it looks like the method could be affecting the results the authors derive later about species overlap, edge distribution inside and across communities etc.

Clustering: I think that the robustness analysis the authors have performed suffices to show robustness of their results given the biogeographical areas they identify.

Analysis of nestedness: I am not sure I understand the analysis of nestedness in the manuscript. The manuscript claims that species dissimilarities across biogeographical regions are often due to nestedness rather than species turnover using the

\beta_nes from Baselga 2010. While the analysis is clear, it is unclear to me whether there is a significantly nested structure in the overall network. That is to say, shouldn't the network be nested for this to be a component in species dissimilarity across regions? Maybe I am missing something here, but it looks to me that in order for the claim of the authors to be fully justified, there should be a nested pattern in the network at least within a biogeographical region, shouldn't it? What I have in mind is at least similar in spirit to some studies that have reported a balance between nestedness and community structure in bipartite networks within ecology and outside ecology, which I think could be also possible descriptions for this data see comment above about patterns that can describe the network (Solé-Ribalta et al PRE 2008).

Version 2:

Decision Letter:

4th February 2025

Dear Dr Bernardo-Madrid,

Thank you for your patience while your manuscript entitled "A general rule on the organization of biodiversity on Earth's biogeographical regions" was under review. As you will see from the reports below, Reviewer 1 has no major remaining points but Reviewer 4 still finds one of their main concerns not satisfactorily addressed. While we are aware that this is already the third round of external review and we wish to avoid further unnecessary iterations, my editorial colleagues and I deem Reviewer 4's criticisms persuasive and must urge you to follow their recommendation to address the issue. We will therefore need to see your responses to the reviewers' comments along with a revised manuscript before we can reach a final decision regarding publication.

* Please highlight all changes in the manuscript text file and provide it in Microsoft Word format, with line numbers.

* If you have not done so already please begin to revise your manuscript so that it conforms to our Article format instructions at <http://www.nature.com/natecolevol/info/final-submission>. Refer also to any guidelines provided in this letter.

* Extended Data Figures - please ensure that any supplementary figures and tables that are crucial to the manuscript's conclusions are converted into Extended Data figures and tables to increase visibility of these data. Extended Data figures and tables are online-only (present in the online PDF and full-text HTML versions of the paper), peer-reviewed display items that provide essential background to the article but are not included in the main article due to space constraints. A maximum of ten Extended Data display items (figures and tables) is permitted.

Link Redacted

Nature Ecology & Evolution is committed to improving transparency in authorship. As part of our efforts in this direction, we are now requesting that all authors identified as 'corresponding author' on published papers create and link their Open Researcher and Contributor Identifier (ORCID) with their account on the Manuscript Tracking System (MTS), prior to acceptance. ORCID helps the scientific community achieve unambiguous attribution of all scholarly contributions. You can create and link your ORCID from the home page of the MTS by clicking on 'Modify my Springer Nature account'. For more information please visit www.springernature.com/orcid.

We look forward to seeing the revised manuscript, preferably within the next eight weeks. If you cannot send it within this time, please let us know. We will be happy to consider your revision so long as nothing similar has been accepted for publication at Nature Ecology & Evolution or published elsewhere.

[redacted]

Reviewers' comments:

Reviewer #1 (Remarks to the Author):

The authors performed a considerable work to improve their manuscript, and I appreciated the efforts performed. I think the present version is much clearer than the previous ones.

I only have a few very minor suggestions to further improve clarity

I think "characteristic / non characteristic species" should be clearly defined at the beginning of the paper

L 257: perhaps a question mark is missing here

L 283: to be honest, in most of cases pool size is more important than sorting, and in several cases pool size is much more important than sorting (eg for amphibians and reptiles you cannot write that the effect of sorting is comparable to the one of pool). Please re-shape the following sentences accordingly

L 319: biodiversity within biogeographical regions?

Reviewer #4 (Remarks to the Author):

First of all, I would like to thank the authors for having addressed all of my comments.

I think that robustness analysis regarding the different data sources used are satisfactory.

However, with respect to the method used to define biogeographical regions, the answer that I have received is not so satisfactory. My feeling is that you have not fully addressed my concern. What I asked was, if I were to use a method that is not 'specifically' for community detection but considers other connectivity patterns (including the modular one you are using), will I find the same patterns that you find?

The answer that you have given me is, in short, that you do not think these other approaches are appropriate, but I disagree. I fully understand that there is literature that uses community detection algorithms to define biogeographical regions (Vialha and Antonelli 2015 and other references). However, that does not mean that this is the best way to model the large scale organization of the network of species abundance in habitats. In my book, the fact that other people do it like this does not mean I should. If there is a possibility that other patterns could explain the data it would still be an interesting finding worth talking about, hence my honest concern. I believe, that I asked a very direct question that is relevant to your findings, and the only way to answer it is to run the stochastic block model based algorithm I suggested using the software developed by T Peixoto (which by the way has been demonstrated theoretically to lower the resolution limit if you use a hierarchical prior) and see what you find. Scientifically, this is the right thing to do because I do not think that you can argue that methods that intrinsically consider bipartite graphs and the best way to summarize the information in those graphs are a priori worse than the approach that you are using.

In all honesty, I know that you have done a lot of work but I also think that you need to fully address all of my concerns before I can accept. Making this very simple experiment and explaining the results are truly going to be good for the ecological community in general. As a final point, these approaches have also been used by ecologists in the analysis of other types of ecological data successfully (for instance see the work of S Kéfi and collaborators) so it is not foreign to the community.

Version 3:

Decision Letter:

13th March 2025

Dear Dr. Bernardo-Madrid,

Thank you for submitting your revised manuscript "A general rule on the organization of biodiversity on Earth's biogeographical regions" (NATECOLEVOL-24020466C). On the basis of Reviewer 4's feedback, we will be happy in principle to publish it in Nature Ecology & Evolution, pending minor revisions to comply with our editorial and formatting guidelines.

[redacted]

Reviewer #4 (Remarks to the Author):

This is a lovely result and a great paper. I congratulate the authors on their work and I deeply appreciate their diligence.

Response to reviewers' comments:

We sincerely thank both reviewers for their positive feedback and kind words. We have made the suggested changes, which are detailed point by point below. However, we would like to note that, in addition to these, we have added the following improvements to the manuscript:

- (1) We conducted new sensitivity analyses by reanalyzing all our analyses using two to eight biogeographical sectors, previously called biogeographical roles. We classified the grid cells of all taxa into two to eight clusters (seven scenarios). We selected this range of clusters because it reflects the optimum number of taxon-specific sectors observed across taxa (old Supplementary Table 9). For each scenario, we represented the distribution of the four biodiversity aspects in each sector (Supplementary Figs. 8-14), mapped the sectors geographically showing the core-to-transition organization (Supplementary Figs. 8-14), performed multinomial analyses that show that the geographical areas defined by the biogeographical sectors exhibit distinct environmental conditions (Supplementary Table 9), and performed nestedness analyses that show that the taxonomic dissimilarities in the areas defined by the biogeographical sectors are consistently associated with a nestedness component (Supplementary Table 10).
- (2) We implemented the calculation of these four aspects of biodiversity in the online tool Infomap Bioregion.
- (3) We enhanced the figures throughout the manuscript with more detailed silhouettes.

Please see below for the specific implementations suggested in each comment.

Rubén Bernardo-Madrid on behalf of all co-authors

Reviewer #1 (Remarks to the Author):

This manuscript combines biogeographical analyses of multiple taxa (plants, vertebrates and dragonflies) to identify generalities in the biogeographical organization of species distribution on Earth.

The key message of the manuscript is that the authors found a rather general pattern that can be defined as a transition from “core” areas to transition areas.

This idea is somehow implicit to many classical biogeographical analyses (see Kreft, H., and W. Jetz. 2013. Comment on "An Update of Wallace's Zoogeographic Regions of the World". *Science* 341:343; see also previous work such as Müller, P. 1974. *Aspects of Biogeography*. Dr. W. Junk b.v. Publishers, The Hague. Müller, P. (1986). *Biogeography*. Harper & Row). What is novel, is the attempt to retrieve these pattern from quantitative biogeographical analyses.

I appreciated several aspects of this manuscript, including the multi-taxa approach and the large number of analyses performed to support the conclusions.

Nevertheless, the manuscript is not entirely clear. There are several points where it is not clear what the authors are actually doing, and why. In some cases this made it difficult assessing the suitability of the manuscript (i.e. in some cases it was not clear if the authors used appropriate methods / followed a strategy that is appropriate to achieve its aims).

In my view, several clarifications are pivotal to make the manuscript fully understandable

We appreciate the reviewer's recognition of the novelty of empirically demonstrating the core-to-transition concept, which we agree that it has been implicit in the mind of biogeographers for centuries (updated text: L. 293-294). We also value the acknowledgment of the effort involved in evaluating multiple taxa and the extensive analyses conducted to support our conclusions. Finally, we thank the reviewer for the constructive feedback, which has allowed us to convey our message better. Please see below our response for each specific comment.

Specific comments

In some cases the manuscript exaggerates its generality. For instance, the title (A general rule on the organization of biodiversity on Earth) is too vague. It seems that you have found a general rule for all biological sciences. It must be clear that the manuscript focuses on biogeography and on species distributions / biogeographical regions

We agree that the title should reflect the focus on biogeographical regions. Accordingly, we rewrote the title:

"A general rule on the organization of biodiversity on Earth's biogeographical regions"

The first paragraph of the introduction is very clear

Thank you for your positive feedback.

L 35: this sentence is a bit awkward, please rephrase

We have changed the sentence:

"Species biodiversity can be described using complementary aspects." (L. 54)

L 36: It is increasingly recognized that no bioregion is "virtually devoid" of life forms. Even Antarctica is increasingly recognized as an environment hosting a significant biodiversity

We have changed the sentence:

"For instance, geographical areas may teem with species or harbour only a few" (L. 54-55)

L 43 (and other parts of the manuscript): here it is quite unclear. Are you focusing on biogeographical or on ecological organization?

Added the term "biogeographical" (L. 63): *"biogeographical organization of biodiversity"*

Results:

The identification of the 7 "roles" is the key result of the manuscript, but note that the generality is at least partially related to the fact that the authors force the 7 roles for all the taxa (in fact the optimal number of roles can be quite different across taxa, see Supplementary Table 9)

Thank you very much for highlighting this crucial point for ensuring the reliability and significance of our findings. Firstly, following the recommendations of Reviewer #3, we have replaced the term "biogeographical roles". Now we refer to "biogeographical sectors". As we offered to reviewer #3, we would be delighted to hear your opinion on this new terminology and any other suggestions you might have.

Regarding the comment on the key result, we apologize for any confusion caused by our writing. For us, our key findings are twofold: first, the existence of non-linear but still gradients in four biodiversity metrics, conceptualized as a core-to-transition organization. Second, the areas represented by these biogeographical sectors are exposed to distinct environmental conditions, and their taxonomic dissimilarities are mainly associated with nestedness; empirically supporting the role of environmental filters as drivers of regional biodiversity. For us, the biogeographical

sectors serve as tools to observe and understand these biodiversity patterns and evaluate their underlying drivers, with the number seven being an optimization. In our previous version, we indicated that the exact number of sectors is not critical for understanding the core-to-transition organization (old version: L.100-110, L. 436-438, and Supplementary Fig. 21). However, this essential information was not explicitly or adequately visible in the main text. In our updated version, we conducted new analyses to demonstrate this fact better. We repeated all our analyses using two to eight general biogeographical sectors (i.e. we classified the grid cells of all taxa into two to eight clusters; seven scenarios). We selected this range of clusters because it reflects the optimum number of taxon-specific sectors observed across taxa (old Supplementary Table 9). For each scenario, we represented the distribution of the four biodiversity metrics in each sector (Supplementary Figs. 8-14), mapped the sectors geographically (Supplementary Figs. 8-14), performed multinomial analyses that show that the geographical areas defined by the biogeographical sectors exhibit distinct environmental conditions (Supplementary Table 9), and performed nestedness analyses that show that the taxonomic dissimilarities in the areas defined by the biogeographical sectors are consistently associated with a nestedness component (Supplementary Table 10). The consistency of the results shows that the number of biogeographical sectors is not critical for our biogeographical inferences. We have specified these new sensitivity analyses in the main text:

L. 133-135: *“Sensitivity analyses using two to eight, instead of seven, general biogeographical sectors also show the core-to-transition organization and the gradient in the four-biodiversity aspects (Supplementary Figs. 8-13).”*

L. 208-213: *“Sensitivity analyses using two to eight general biogeographical sectors showed that all our results were robust to methodological aspects such as the choice of the optimal number of clusters to describe biodiversity within biogeographical regions (i.e., biogeographical sectors common across taxa, the existence of a core-to-transition organization, the association of the sectors with distinct environmental conditions, and the taxonomic dissimilarity between sectors mostly driven by nestedness; Supplementary Figs. 8-14 and Supplementary Tables 9 and 10). The congruency between our predictions and results supports the important and general role of environmental filters acting on both characteristic and non-characteristic species in shaping biodiversity in biogeographical regions”*

We acknowledge that our previous comparison between the piecewise-regression criterion (an elbow-like method) and the Calinski-Harabasz criterion could raise some doubts about the robustness of our biogeographical inferences; especially because we forgot to indicate that the general number of clusters was seven when using both criteria. In this new version, we simplified the study by removing the sensitivity analyses of Calinski-Harabasz, which could suggest that the optimal number is another. Note that by repeating all our analyses using a distinct number of clusters, we reduce the importance of choosing a cut-off point, and thus the criterion used. We hope that these new analyses help readers see that methodological decisions are not critical for observing the core-to-transition pattern and learning about its drivers (Supplementary Figures 8-14 and Supplementary Tables 9 and 10).

the interpretation of the 7 “roles” should be explained in the text, or in a clearer version of Fig 1 (or perhaps both...). What do you mean eg by role 1? (eg role 1 = high endemcity, low biota overlap, high relative species richness, low average occupancy...). I understand that writing

everything in the text can make it the results length, but the authors should give to readers a key for a faster interpretation of Fig 1A (at least to me, understanding these roles was not very easy).

We appreciate the reviewer's constructive feedback. We agree on the difficulty of including such detailed information in the main text and following the suggestion we have provided more detail in Fig. 1, where this information is better conveyed through the visual representation of the histogram and maps.

While we can understand the value of indicating high, medium, or low levels for each metric within each sector, we believe that graphical representations like the histograms allow readers to understand the distribution and relative values of each metric quickly and better, eliminating the need for lengthy textual explanations. Still, we totally agree on the need to further aid in the biogeographical interpretation of the sectors. In our new version, we have added a new legend in Fig. 1D and we have updated the caption of Fig. 1. We believe these changes make the interpretation of the sectors clearer, nevertheless we are open to further suggestions.

Fig. 1 | Seven general and spatially structured biogeographical sectors characterized from four biodiversity aspects across diverse taxa. (A) Biodiversity value distributions observed within each of the seven general biogeographical sectors. (B) Two main axes of a principal component analysis of the grid cells of the seven taxa, explaining 76.31% of the variance. (C) Frequency (represented by the size of the circle) of neighbouring probability above that expected by chance among biogeographical sectors from all biogeographical regions and taxa (test of proportions of neighbours' grid cells for each biogeographical region; p -values < 0.05). (D-G) Examples of the spatial distribution of the biogeographical sectors in birds (D), trees (E), dragonflies (F), and rays (G). In Fig. 1D-F, warm colours (reds) represent biogeographical sectors associated with relatively low overlap of biotas, while cool colours (blues) indicate areas with higher overlap, whose relative extension across the surface of the biogeographical regions may support the existence of low and high biota overlap regions³⁰. Darker shades in both colour groups represent higher richness of characteristic species. Dark red and blue are the areas of the biogeographical region with the highest richness of characteristic species, the lowest biota overlap, the highest endemism, and the lowest widespread species; what we called the most core areas of the low and high biota overlap regions (Fig. 1D). A shift between adjacent sectors reflects a decrease or increase in two or more biodiversity aspects. However, from the most core areas to the most transitional areas of each region, a non-continuous gradient is observed across all four aspects. Biogeographical regions in trees (E) and rays (F) have been manually edited to facilitate visualization (original biogeographical regions and biogeographical sectors for all taxa are shown in Extended Data Figs. 2-8).

Fig 1 is central to the understanding of the manuscript

I do not think the scale is appropriate. This color scheme gives the impression of a linear gradient, but we do not have a continuous gradient. I suggest using a discrete color scheme that gives more the idea of clusters. In fact in the text (eg L 90-99) you nicely represent these roles using rather clear terms (eg "hotspots" vs "transitional areas"). It would be nice associating these terms to the 7 roles. A discrete color scheme allows a better identification of the roles on the map (this would be important in both Fig 1 and Fig 2). Consider approaches such as the one used in the supplementary figures S1-S2 of Holt et al (2013 Science).

NMDS-like approaches might also be used to clarify the interpretation of “roles” along the core-transition organization

We thank the reviewer for all the useful suggestions. We agree that multivariate analyses can provide complementary insights into the relationships among sectors. As such, we have added the results of a PCA to Figure 1 (Panel B).

We also concur with the need to help associate the biogeographical sectors with the biogeographical terms: “core” and “transitional” areas. Therefore, in our new version, we have added a guide in Fig. 1D. Moreover, we added new text in the caption to provide a clearer and quicker biogeographical understanding of the maps (see previous comment).

We fully agree that sectors do not represent a single linear gradient. However, the sectors do reflect groups defined by continuous variables, where each shift between sectors involves a decrease or increase in the values of two or more biodiversity aspects. Moreover, the shift between the most core and most transitional sectors involves a decrease or increase in the values of all four biodiversity aspects. The existence of this non-linear gradient, described as the core-to-transition organization, is a key result and the clue to evaluate the mechanisms shaping biodiversity within biogeographical regions (L. 177-178). For us, this message would be harder to understand using a more discrete colour scheme (see example below). Therefore, to reflect this non-linear gradient, we prefer to keep the current colour scheme and explicitly indicate its non-linearity in the legend (see response to previous comment).

Please note that the reason why we used two main colours—warmer and cool—is to reflect biota overlap and highlight two broad “types” of high and low overlap biota regions supported in the literature. We also used the shades to primarily reflect species richness but also gradients in occupancy and endemism.

We believe that the explanation of the association between colours and biodiversity aspects (Legend Fig. 1), together with our guide (Fig. 1D) and the reviewer’s suggestion (Fig. 1B), helps to understand the biogeographical meaning of the figures more easily and quickly.

L 104: see also Kreft, H., and W. Jetz. 2013. Comment on "An Update of Wallace's Zoogeographic Regions of the World". Science 341:343 and references therein

Added

L 165: Please be more explicit about your predictions
Changed (L. 205-208).

“However, most results across biogeographical regions and taxa align with our predictions about different environmental conditions associated with distinct biogeographical sectors and nestedness being the key driver of their biotic dissimilarity.”

L 173-174: Is 90% more than what would be expected by chance? This can be tested by creating “null” core areas having the same average size of the true core areas and calculating the % hosted species (and repeating this a number of times).

Thank you for the constructive feedback. This information could be valuable for conservation planning. We have performed these null models for each biogeographical region and added the results to the main text (L. 220-225).

“This hypothesis²⁶ is supported by the fact that the most core areas of each biogeographical region cover approximately 30% of the region’s surface, harbouring more species than the remaining 70% area (holding ca. 90% of species, a species richness higher than expected by chance based on their geographical extension in reptiles, amphibians, dragonflies and trees; p-values < 0.05 in models permuting sectors identity).”

Paragraph “influence across spatial scales”

In general, this is the less clear paragraph of the manuscript. The rationale of the analyses is not very clear to me and should be better explained. A key result is that the (standardized) number of characteristic species is a good predictor of the total species richness of a cell. Is this surprising? If I correctly interpreted this analysis, this seems to be a quite obvious expectation.

Furthermore, from the text (and the methods) I had the impression that the 3 variables were included jointly in the same analysis as independent, but I guess this is not the case, as the summed % explained variance (pool size + characteristic + non-characteristic species) sometimes reach values close to 100%

We apologize for the lack of clarity in our previous text. We acknowledge that in our previous version, our goals—understanding the relative importance—as well as the previous knowledge about the empirical influence of these factors, and the contrasting meaning of their different dominance were not sufficiently clear, making the novelty of the question and the approach to address it difficult to appreciate. We have therefore rewritten the section and added new information in the main text and methods.

To clarify the rationale of the analyses in our new version, we have explained how the variation in local species richness can be affected by two regional effects: the size of the species pool and the sorting of species causing the core-to-transition organization (L. 248-253), and that our goal is to understand their relative importance (L. 253-254). We have explicitly indicated that to quantify their relative influence, we approximately decomposed the observed species richness variability into proxies of the regional species pool and of the regional species sorting (L. 261-269; L. 632-644), and then evaluated their relative influence by using variance partitioning (L. 272-274 and L.654-660). As the three explanatory variables approximately represent a decomposition of the observed species richness variability at the grid cell level, together they should explain a large fraction of the variance in species richness (L. 641-642). However, as more explicitly detailed in our new version, we are interested in determining which factor is more important in relative terms: the size of the regional species pool, or the sorting of characteristic and non-characteristic species (L. 269-272, 643-644). In this context, if there are large differences in the pool size across biogeographic regions and the sorting of regional pools only attained few species, the pool size should capture more variance in species richness than the sorting of regional species pools, and vice versa (L. 644-653). Thus, a higher explanatory capacity of the richness of characteristic species than that of the pool size should not be obvious. To our knowledge, while the importance of the size of the regional species pool in explaining local variations in richness across the planet

has received considerable attention (e.g. Ricklefs and He. 2016.PNAS,113,674–679), the empirical importance of species sorting within the biogeographical regions on a global scale is more uncertain. Our results contribute to fill this gap, linking the core-to-transition organization with global patterns of species richness.

We have also rewritten the methods section to make the aforementioned points clearer (L. 629-665)

L 197: the variance of what? Please better explain the dependent and independent variables Specified

“To address the relative importance of the core-to-transition organization in shaping global patterns in species richness, we modelled the variance in species richness across grid cells using three variables” (L. 261-263).

L 211-212: note that you did not test if this holds across all the bioregions – you calculated average values across all the bioregions, but it is possible that this holds for some bioregions and not for other bioregions

We indicated its “on average” signal.

“Linear regressions and variance partitioning showed that the combined influence of species sorting can be on average comparable or more important than the size of the regional species pool” (L272-274)

“Thus, the processes underlying species sorting within biogeographical regions, likely tied to regional environmental filters and responsible for core-to-transition organization, may be, on average, as important as variations in regional pool size driven by the balance of speciation, extinction, and biogeographical dispersal” (L. 276-279)

L 212-213: this sentence needs references

With the rewrite, we have removed that sentence.

Conclusive paragraph:

L 233-235: this is the reason why biodiversity hotspots were originally defined (Myers et al 2000. Biodiversity hotspots for conservation priorities. Nature 403:853-858)

Reference added.

L 356-357: this explanation of non-characteristic species is too long and makes it the sentence unclear. I suggest explaining well the definition of non-characteristic species above (around L 346-348), and here just using the term “non characteristic species”

We agree and we have shifted the information.

“Species present in a grid cell of a biogeographical region but not assigned to the same module were considered as non-characteristic species (Extended Data Fig. 1). Non-characteristic species can be more affined to another biogeographical region—clustered in another module—or not affined to a particular biogeographical region, clustered alone in a module.” (L. 443-447)

Formulas used to calculate the 4 metrics should be explicit in Fig S1.

For instance, in Fig S1-step 5 - relative species richness, it should be clear that it is calculated as

(cell species richness – mean(SR of the bioregion))/ (SD(SR of the bioregion))

And so on.

Elsewhere it is very tricky to understand where do 1, 2, 0.57... come from

Fig S1 it is quite important to understand the meaning of the 4 metrics (particularly of occupancy).

Consider improving this figure, and moving it to the main text

Thanks for the constructive feedback. We agree and we have updated Extended Figure 1.

L 402; this should be Supplementary Tables 1-8. Also note that the absolute value of r should be the relevant value here

We have moved to Supplementary Tables 1-8 and indicated that we used the absolute value.

Thanks for pointing out this error in the mathematical formulation.

“The four biodiversity aspects are not strongly correlated ($|$ Pearson’s coefficient $| < 0.7$; ⁶⁰), and thus provided complementary information (see correlation values in Supplementary Tables 1-8).” (L. 500-502)

L 409: tiny biogeographical regions...

Added

L 413-416: Please explain why you used this approach, and not alternative approaches to the selection of the number of clusters. Consider showing the plots of goodness-of-fit and the detected thresholds in a supplementary figure

We agree that in our previous version, the goodness-of-fit would have been helpful for readers to understand the robustness of our results to methodological choices. However, in our new version, with the sensitivity analyses repeating all analyses using seven different numbers of clusters—ranging from two to eight clusters (Supplementary Figs. 8–14 and Supplementary Tables 9 and 10)—this information may no longer be necessary. Therefore, given our new sensitivity analyses, we prefer not to include this information for the sake of simplicity, though we remain open to further suggestions on this matter.

As for the criterion, we have used an elbow-like method since it is commonly used in biogeography.

“We determined the optimal number of clusters or biogeographical sectors per taxonomic group by using an elbow-like method⁶¹ commonly used in biogeography^{62,63} (see sensitivity analyses below)”. (L. 510-511)

L 421: to be honest, the Calinski-Harabasz index produced rather different results (eg Amphibians: piecewise regression 8, Calinski-Harabasz index 2; Rays: 8 vs. 3)

Also in this case, showing the plots may help to evaluate the robustness of selected thresholds

We agree that the optimal number of taxon-specific sectors varies depending on the criterion used. However, when identifying the number of general sectors, using the taxon-specific sectors determined by either piecewise regression or the Calinski-Harabasz index as references, we consistently found seven general sectors as the optimal solution. This was not adequately explained in our previous version.

Nevertheless, as indicated previously, we removed references to the Calinski-Harabasz index in the new version for the sake of simplicity. Instead, we replicated all analyses using two to eight general sectors (Supplementary Figs. 8-14 and Supplementary Tables 9 and 10), providing

stronger evidence of the robustness of our results and their independence from clustering thresholds.

L 422-423: this is really unclear to me. How did you cluster all cells together for the global analysis? The spatial extent (and even the grain) is very different across taxa

We consider each cell as a unit in the clustering analyses ignoring the differences in extent and grain. The identification of general sectors using data with distinct extent, grain and collection sampling support that our results are robust to the type of the data, and thus the biological reality of our inferences (L. 78-80).

L 431: please add references for the AMI. I acknowledge that I'm not an expert of this approach
Done.

L 511: here do you mean dissimilarity between cells within bioregion? Or between "roles" within bioregions?

We have clarified that it is between biogeographical sectors.

"We examined whether the variation in species composition among biogeographical sectors, measured by Sørensen pairwise dissimilarity, was more attributed to nestedness or to species turnover components³⁸" (L. 615-617)

"For each bioregion and taxon, considering all present biogeographical sectors together, we computed the proportion of the taxonomic dissimilarity between sectors attributed to nestedness." (L. 620-621)

Reviewer #3 (Remarks to the Author):

I find that the manuscript represents a very interesting work. I will not comment on the methods, but on a few issues that I think may help improve the manuscript.

Thank you for your kind words regarding the manuscript. We appreciate your expertise, and your research has been an inspiration that led us to ask the questions presented in this study. We would be grateful for any comments you might have to improve our study, and especially for your opinion on how we should term the new biogeographical entity we are describing.

“Non-characteristic species” (e.g., lines 67, 72, etc.): I am not sure if this means non-endemic species? This should be clarified.

We clarified in the text that characteristic species can have parts of their distribution range outside their bioregion and thus be non-endemic. That is, both characteristic and non-characteristic species may be non-endemic. In our revised version, we have clarified their meaning:

Main text (L. 83-89)

“This algorithm identifies modules of highly connected grid cells and species^{18,24}, where grid cells included in a module form the biogeographical regions (Extended Data Figs. 2-8), while species in a module represented the characteristic species of that biogeographical region¹⁸, which are those with all or most of their distribution range in that biogeographical region. Species occurring in the grid cells of a biogeographical region but not grouped in the same module are present but non-characteristic species¹⁸.”

Methods (L. 439-447)

“The output of network analyses is sets of highly connected grid cells and species, referred to as modules. For each module, we considered the grid cells as the geographical areas of a biogeographical region, and the species as its characteristic species pool. Characteristic species have their entire distribution range or most of it in their associated biogeographical region (thus including endemic, but also non-endemic species). Species present in a grid cell of a biogeographical region but not assigned to the same module were considered as non-characteristic species (Extended Data Fig. 1). Non-characteristic species can be more affined to another biogeographical region—clustered in another module—or not affined to a particular biogeographical region, clustered alone in a module.”

“Roles” (e.g., lines 78, etc.): this word seems a little awkward to me. I suggest the authors to provide a better one.

We agree that "roles" may not be the most appropriate, as a geographical area is not "playing a role" in the literal sense. In our revised version, we propose "sectors" to indicate distinct compartments or areas with potentially different properties. We are open to other suggestions.

I find that the “Extended Data Figures 2-8” represent a great illustration of the results, and I don’t think it is appropriate to have them relegated to an appendix. I suggest the authors to incorporate them to the main text.

Thank you for your valuable suggestion. We agree that the “Extended Data Figures 2-8” significantly enhance the presentation of our results and have now moved part of them to the main text (Figs 1 and 2).

Juan J. Morrone

Reviewers' comments:

Reviewer #1 (Remarks to the Author):

The authors performed a considerable work to improve their manuscript, and I appreciated the efforts to perform new analyses (eg the sensitivity analysis). The introductory and conclusive part, as well as result interpretation, are generally clear.

Nevertheless there are a few points where some clarifications are needed

Thank you very much for the positive comments. Please see below our response to each comment.

For me, the most confusing aspect of the manuscript is how the results from different taxa (different areas, different resolution...) were merged using clustering and AMI. This is explained in the methods (please note that I'm not an expert of these approaches; which values are used here?). However, I think that the main text (eg par "general organization of biodiversity") requires a short description of the fact that maps of multiple taxa were summarized to produce results in Fig 1 A-C

We agree with the reviewer and we have included a concise explanation in the main text of why and how we arrived at Fig. 1A–C:

Lines 114-120: "To interpret the biogeographical meaning of sectors, we analyzed the distribution of biodiversity aspects across the seven sectors using values from all grid cells of the seven taxa (Fig. 1A). We also assessed the similarity of sectors in a multidimensional space defined by four biodiversity metrics using Principal Component Analysis (Fig. 1B). We finally mapped sectors and, for each taxon and biogeographical region, assessed whether neighbouring relationships between sector pairs occurred more frequently than expected by chance (binomial proportion tests, p-values < 0.05; Fig. 1C)."

Complementarily, we have provided more information in the caption of Figure 1. On the other hand, we are unsure which specific values the reviewer is referring to with the comment, "which values are used here." However, we acknowledge that our previous text was unclear and, in our updated text, we have provided more detailed explanations of the methods, particularly the clustering and AMI steps (please see our updated text at L. 548-588).

L 101-103: this sentence is not very clear to me. What do you mean by "one taxon"? do you refer to the comparison between the analyses performed on different "taxa" (eg birds vs amphibians) on the same cells? Or something different?

By one taxon we mean one of the seven taxonomic groups we analysed (eg birds, amphibians).

We have revised the text for clarity:

If biodiversity is organized differently across taxa, grid cells for each of the seven studied life-forms should be clustered separately from each other. Conversely, if

biodiversity is organized similarly across taxonomic groups, clusters would combine grid cells from all taxa. (L. 106-108).

Regarding the second question, our analysis does not involve statistical comparisons between taxa beyond using k-means clustering with all grid cells from the seven taxa. Each taxon was projected onto its own grid, resulting in a distinct set of grid cells for each taxon. We performed a k-means clustering analyses using all grid cells of the seven taxa. Clustering was exclusively based on the similarity of biodiversity aspects, without consideration of taxon membership, grid resolution, or geographical location. We apologize for our previous unclear text and have clarified these details in the updated main text and methods (Lines 94-101, 450-452, 465-588).

L 105: Which result? I'm missing a sentence assessing the similarity between taxa

We have explicitly detailed the results and highlighted the similarity between taxa, as evidenced by the generality of the sectors (clusters formed by grid cells from all taxa; please see details in our previous response).

The generality and low number of biogeographical sectors across the seven taxa support the hypothesis that biodiversity at regional scales is arranged in a consistent and limited number of ways, and that the mechanisms governing regional biodiversity transcend the particularities of individual life forms (L. 110-113)

Fig 1C reports an interesting conclusion that can be more explicitly reported in the text. What is the proportion of significant tests?

Thank you for this constructive suggestion. Since we conducted 36 pairwise analyses (total combinations of sector pairs) and defined sectors by colors, we would prefer to provide this information in a table format in a new Supplementary Table 9. However, if the reviewer has specific suggestions for addressing this complexity in the text, we are open to incorporating them.

Supplementary Table 9 | Neighbour Analysis. *The area below the diagonal shows the total number of biogeographical regions, across the seven taxa, where the two compared biogeographical sectors were represented (i.e., at least one grid cell from each of the two sectors). The area above the diagonal displays the proportion of those biogeographical regions where neighbouring between pairs of sectors was higher than expected (expected probability = 1/6). Statistically significant adjacencies between sectors were identified using bidirectional binomial proportion tests for each pair of sectors (A→B and B→A), comparing observed neighbouring frequencies to expected probabilities under random distribution ($p < 0.05$). Fig. 1C shows the relative frequency of statistically significant neighboring events between pairs of biogeographical sectors across all bioregions and taxa, normalized by the total number of significant events..*

-	0.84	0.06	0.25	0.00	0.00	0.57
83	-	0.71	0.69	0.05	0.04	0.24
54	55	-	0.58	0.42	0.00	0.02
95	81	53	-	0.55	0.16	0.41
94	75	50	86	-	0.45	0.54
31	25	12	31	44	-	0.43
92	70	48	90	93	46	-

Multinomial models suggest that environmental features (eg precipitation and temperature) explain a good proportion of the assignment to biogeographical sectors within bioregion.

In principle, this can be also caused by spatial autocorrelation (both climate and species distribution are affected by strong spatial autocorrelation). Are these results consistent if you take into account spatial autocorrelation into your models? The same issue applies to the use of past climate.

Spatial autocorrelation may be relevant also for the linear models described at L 654 (part of variation is probably just explained by the spatial component)

We thank the reviewer for highlighting this point. The implications of spatial autocorrelation remain a debated topic in ecology (Legendre 1993; Lennon 2000; Diniz-Filho et al. 2003; Hawkins et al. 2007; Diniz-Filho et al. 2007; Bini et al. 2009; Hawkins 2012; Kühn & Dormann 2012). We conducted our sensitivity analysis based on the perspective of Diniz-Filho et al. (2003), Bini et al. (2009), Hawkins (2012), and Kühn and Dormann (2012), which can be summarised with this statement:

“If the spatial autocorrelation of an ecological response variable is caused by autocorrelated predictor variables (such as climate, land use, topography, human population densities or virtually any other spatial predictor), we are not alarmed. Of course we do not wish to remove this effect of such predictors. When all relevant predictor variables are included in non-spatial models, the residuals will not be autocorrelated. Under these rare conditions autocorrelation is neither an artefact nor a problem. SA in the residuals is, however, a serious problem” (Kühn and Dormann 2012)

We differentiate autocorrelation as a pattern in nature versus autocorrelation in residuals of statistical models (Bini et al. 2009; Hawkins 2012), and that the latter may lead to interpretation problems when focusing on significance levels—Type I error—rather than the explanatory power of variables or the relationships between response and explanatory variables measure by the raw (conventional) or standardized regressions coefficients (Diniz-Filho et al. 2007; Bini et al. 2009; Hawkins 2012). Therefore, our new sensitivity analyses focused on evaluating residual spatial autocorrelation, addressing it when present, and assessing whether the environmental signal persists. Our results show that the environmental signals persist across all biogeographical regions, except for one bird region, reinforcing the robustness of our inferences. These findings are discussed in the main text and methods. To perform the sensitivity analyses, we consulted with Dr. Eric Gerber, expert in the estimation of residuals in multinomial models (Gerber & Craig 2024) because no established scientific method exists for estimating residuals in our specific multinomial models. We detailed our analyses in a new Appendix B.

Regarding the reviewer's comment: "*Spatial autocorrelation may be relevant also for the linear models described at L 654 (part of the variation is probably just explained by the spatial component),*" we agree with Diniz-Filho et al. (2003) that controlling for macro-scale autocorrelation should only be done if there is reason to believe that macro-scale correlations between explanatory factors and species richness are spurious, and that other processes, acting at smaller spatial scales, are the ultimate drivers of species richness. However, in our case, we model species richness using variables derived directly from its decomposition, ensuring that these relationships are not spurious and precluding the influence of alternative factors. The observed spatial correlation is both real and inevitable, as it reflects the inherent structure of species richness itself and its decomposition. Our primary objective is to identify which decomposed variable shows the strongest correlation with species richness; a correlation that does not constitute an analytical issue but our research target (Diniz-Filho et al. 2003; Bini et al. 2009; Hawkins 2012; Kühn & Dormann 2012). In summary, the main motivation to correct for spatial autocorrelation (i.e., to avoid spurious relationships) is not present. Moreover, including spatial components in models of decomposed variables could obscure the natural patterns we aim to investigate (Bini et al. 2009; Hawkins 2012). For these reasons, unlike in the multinomial models, we chose not to include spatial elements. However, if the reviewer has further specific concerns about how spatial autocorrelation might affect these results, we are open to conducting additional analyses based on their guidance.

Minor comments:

Fig 2: the inset is too small.

We agree and have aimed to increase it. However, it remains small, and have considered an alternative inset to illustrate the meaning of the colors or biogeographical sectors, also based on Fig. 1.

Reviewer #1 references

- Bini, L.M., Diniz-Filho, J. A. F., Rangel, T. F., Akre, T. S., Albaladejo, R. G., Albuquerque, F. S., ... & Hawkins, B. A. (2009). Coefficient shifts in geographical ecology: an empirical evaluation of spatial and non-spatial regression. *Ecography*, 32(2), 193-204.
- Diniz-Filho, J. A. F., Bini, L. M., & Hawkins, B. A. (2003). Spatial autocorrelation and red herrings in geographical ecology. *Global ecology and Biogeography*, 12(1), 53-64.
- Gerber, E. A., & Craig, B. A. (2024). Residuals and diagnostics for multinomial regression models. *Statistical Analysis and Data Mining: The ASA Data Science Journal*, 17(1), e11645.
- Hawkins, B. A., Diniz-Filho, J. A. F., Mauricio Bini, L., De Marco, P., & Blackburn, T. M. (2007). Red herrings revisited: spatial autocorrelation and parameter estimation in geographical ecology. *Ecography*, 30(3), 375-384.
- Hawkins, B. A. (2012). Eight (and a half) deadly sins of spatial analysis. *Journal of Biogeography*, 39(1), 1-9.
- Kühn, I., & Dormann, C. F. (2012). Less than eight (and a half) misconceptions of spatial analysis. *Journal of Biogeography*, 39(5), 995-998.
- Legendre, P. (1993). Spatial autocorrelation: trouble or new paradigm?. *Ecology*, 74(6), 1659-1673.
- Lennon, J. J. (2000). Red-shifts and red herrings in geographical ecology. *Ecography*, 23(1), 101-113.
- Wang, K., Abdulah, S., Sun, Y., & Genton, M. G. (2023). Which parameterization of the Matérn covariance function?. *Spatial Statistics*, 58, 100787.

Reviewer #4 (Remarks to the Author):

The authors make an analysis of the organization of biodiversity in biogeographical regions using mostly a networks approach. I agree with one of the reviewers who says that while the ideas presented in the manuscript are not necessarily new, the manuscript offers quantitative evidence for those ideas.

Nonetheless, I have been asked to make a technical assessment on the manuscript, so I will make comments on those technical aspects of the study that I think deserve special attention. That being said, I know that this is at least a second round of review, but I do think that these few points should be addressed before making a final decision on the suitability for publication of this manuscript.

Thank you very much for your comments and suggestions. Please find our responses to each point below.

Data: The authors pool together different datasets corresponding to different types of fauna and even plants. However, not all of the datasets cover the same region worldwide. For instance, the data on trees is only available for North America. My question here is whether the results are affected by the fact that not all data is equally distributed geographically and how that affects the biogeographical regions the authors identify and the metrics the authors define.

We have now evaluated the effect of geographical cover or extent in the definition of bioregions and metrics. For this we focused on the five taxa (amphibians, birds, mammals, rays, and reptiles) with global datasets, for which we could define smaller regions that were still large enough to be biogeographically relevant. In particular, for each taxa we first identified the largest hierarchical level representing biogeographical realms (generally continental extents) and then treated those as if they were separate datasets that we analysed to define biogeographical regions and calculate our metrics. We then compared the biogeographical regions and metrics within realms that we identified when we used the whole global datasets vs those we identified using the smaller, realm-based datasets. Comparisons of bioregions were based on Adjusted Mutual Information (AMI). Comparisons of metric values were based on Spearman's correlations. Overall, identified biogeographic regions and metrics were very consistent regardless of the geographic cover of the dataset used to define them with AMI values very close to 1 (mean \$\pm\$ standard deviation = \$0.94 \pm 0.03\$ ) and very high positive Spearman's rho coefficients (mean \$\pm\$ standard deviation = \$0.83 \pm 0.08\$ ). These results support the robustness of our results at different extents—and the generality of our inferences across the seven taxa analyzed. We have now included this sensitivity analyses as Appendix C.

Obtention of biogeographical regions: To determine 'biogeographical' regions in the bipartite network defined by geographical regions and species, the authors use Infomap, a method that aims at finding groups that contain species and geographical regions together. However, the authors do not justify this choice. In the networks literature, there are many papers that talk about detecting 'communities in bipartite networks' by finding groups of species and groups of geographical areas 'separately' (see at least papers from PRE 2007 – Barber and Guimera et al). I am not sure why the authors did not adopt this methodology instead, since it seems more appropriate for the type of network the authors deal with than the approach used in the

manuscript. Indeed, there are more modern approaches that do not assume that groups of species (or communities) need to comprise species that 'are more likely to be connected to the same groups of geographical areas'. Indeed other connectivity patterns of the bipartite network might explain the patterns we observe, so it seems not as compelling to find these biogeographical regions when using a method that is looking for precisely this type of signal (i.e. densely connected groups of geographical areas and species) that a method that is not (see for instance Gerlach et al Science Adv. 2018 or Cobo-López et al PNAS Nexus 2022). I think it is important for the authors to check the consistency of their results. Right now, it looks like the method could be affecting the results the authors derive later about species overlap, edge distribution inside and across communities etc.

The reviewer is correct in highlighting that we initially omitted an explanation for our choice of Infomap. In the revised manuscript, we have addressed this omission and clarified the rationale behind our methodological approach (L. 480-494). We first emphasized that identifying both bioregions and their characteristic species is essential for calculating biodiversity metrics (e.g. biota overlap). Infomap, which simultaneously classifies grid cells and species within a single integrated framework, avoids the need for secondary steps outside the original analytical framework to assign species pools to bioregions. The reduction of methodological complexity and subjectivity is a major advantage when comparing community detection algorithms and alternative clustering approaches (Vilhena and Antonelli 2014). Second, by explicitly assigning species pools to bioregions, Infomap ensures a biologically meaningful definition of biogeographical regions—areas of Earth's surface characterized by distinct species pools (Nelson, 1978; Rueda et al. 2013; Bernardo-Madrid et al., 2019). Third, we elaborated on the specific advantages of Infomap over other community detection algorithms in the context of our study. These advantages include: its successful application in prior biogeographical exercises across global, continental and regional scales and across multiple taxa (e.g., Vilhena & Antonelli, 2015; Costello et al., 2017; Edler et al. 2017; Bloomfield et al., 2018; Bernardo-Madrid et al., 2019; Calatayud et al., 2019; Yusefi et al., 2019; Fragkopoulou et al., 2024), its use as benchmark of new biogeographical delineation methods (Maestri & Duarte, 2020), as well as its lower resolution limit to identify communities (Smiljanić et al., 2023) and reduced degeneracy compared to other community detection algorithms (Calatayud et al., 2019). This additional information highlights that Infomap provides a robust, well-tested, and widely accepted framework that directly meets the needs of our study by identifying bioregions and their corresponding regional species pools.

We appreciate the reviewer's constructive suggestions. While alternative suggested methods that separately identify modules of grid cells and species might be useful in other contexts, such methods do not align with our current objectives. We believe that this elaboration has strengthened the clarity and depth of the manuscript, and we thank the reviewer again for their valuable input.

Clustering: I think that the robustness analysis the authors have performed suffices to show robustness of their results given the biogeographical areas they identify.

Thank you for affirming our robustness analysis.

Analysis of nestedness: I am not sure I understand the analysis of nestedness in the manuscript. The manuscript claims that species dissimilarities across biogeographical regions are often due to nestedness rather than species turnover using the β_{nes} from Baselga 2010. While the analysis is clear, it is unclear to me whether there is a significantly nested structure in the overall network. That is to say, shouldn't the network be nested for this to be a component in species dissimilarity across regions? Maybe I am missing something here, but it looks to me that in order for the claim of the authors to be fully justified, there should be a nested pattern in the network at least within a biogeographical region, shouldn't it? What I have in mind is at least similar in spirit to some studies that have reported a balance between nestedness and community structure in bipartite networks within ecology and outside ecology, which I think could be also possible descriptions for this data see comment above about patterns that can describe the network (Solé-Ribalta et al PRE 2008).

The reviewer is correct that the overall network should exhibit a certain degree of nestedness across biogeographical regions. However, we apologize for any misunderstanding and wish to clarify that our study does not focus on nestedness across biogeographical regions but rather on nestedness across biogeographical sectors within bioregions. Our analysis aligns precisely with the analyses suggested by the reviewer: *"there should be a nested pattern in the network at least within a biogeographical region, shouldn't it?"*

To better articulate our goals and analyses, we have revised our manuscript for clarity. Please see below for the detailed explanations:

Lines 198-200. *"Differences in species composition among these biogeographical sectors should predominantly arise from one sector's species being a subset of those present in another, representing nestedness patterns rather than species turnover."*

Lines 206-210: *"Complementarily, the partitioning of beta dissimilarity into nestedness and turnover components³⁸ across biogeographical sectors per biogeographical region revealed that taxonomic dissimilarity is more driven by nestedness in 77±2% of biogeographical regions across taxa"*

Lines 668-669: *"we computed the proportion of the taxonomic dissimilarity between sectors mostly attributed to nestedness"*

Lines 663-669: *"We examined whether the variation in species composition among biogeographical sectors, measured by Sørensen pairwise dissimilarity, was more attributed to nestedness or to species turnover components³⁸. In our case, nestedness refers to dissimilarity arising because the species composition in one biogeographical sector is a subset of the species found in another sector, while turnover captures dissimilarity due to species replacement³⁸. For each bioregion and taxon, considering all present biogeographical sectors together, we computed the proportion of the taxonomic dissimilarity between sectors attributed to nestedness."*

Reviewer #4 references

- Bernardo-Madrid, R., Calatayud, J., González-Suárez, M., Rosvall, M., Lucas, P. M., Rueda, M., ... & Revilla, E. (2019). Human activity is altering the world's zoogeographical regions. *Ecology Letters*, 22(8), 1297-1305.
- Bloomfield, N. J., Knerr, N., & Encinas-Viso, F. (2018). A comparison of network and clustering methods to detect biogeographical regions. *Ecography*, 41(1), 1-10.
- Calatayud, J., Bernardo-Madrid, R., Neuman, M., Rojas, A., & Rosvall, M. (2019). Exploring the solution landscape enables more reliable network community detection. *Physical Review E*, 100(5), 052308.
- Costello, M. J., Tsai, P., Wong, P. S., Cheung, A. K. L., Basher, Z., & Chaudhary, C. (2017). Marine biogeographic realms and species endemism. *Nature communications*, 8(1), 1057.
- Edler, D., Guedes, T., Zizka, A., Rosvall, M., & Antonelli, A. (2017). Infomap bioregions: interactive mapping of biogeographical regions from species distributions. *Systematic biology*, 66(2), 197-204.
- Lenormand, M., Papuga, G., Argagnon, O., Soubeyrand, M., De Barros, G., Alleaume, S., & Luque, S. (2019). Biogeographical network analysis of plant species distribution in the Mediterranean region. *Ecology and Evolution*, 9(1), 237-250.
- Maestri, R., & Duarte, L. (2020). Evoregions: Mapping shifts in phylogenetic turnover across biogeographic regions. *Methods in Ecology and Evolution*, 11(12), 1652-1662.
- Nelson, G. (1978). From Candolle to Croizat: comments on the history of biogeography. *Journal of the History of Biology*, 11, 269-305.
- Smiljanić, J., Blöcker, C., Holmgren, A., Edler, D., Neuman, M., & Rosvall, M. (2023). Community detection with the map equation and infomap: Theory and applications. *arXiv preprint arXiv:2311.04036*.
- Vilhena, D. A., & Antonelli, A. (2015). A network approach for identifying and delimiting biogeographical regions. *Nature communications*, 6(1), 6848.
- Yusefi, G. H., Safi, K., & Brito, J. C. (2019). Network-and distance-based methods in bioregionalization processes at regional scale: An application to the terrestrial mammals of Iran. *Journal of Biogeography*, 46(11), 2433-2443.

We sincerely thank the editor and reviewers for their valuable feedback, which has improved our study. In our revised version, we have addressed all requested changes and analyses. In response to Reviewer #1's comments, we have clarified the text accordingly. Following Reviewer #4's suggestions, we conducted new sensitivity analyses using the Stochastic Block Model (SBM) to delineate bioregions, finding strong congruence with our main results. Additionally, we have provided complementary literature-based evidence that further supports the robustness of our biogeographical delineation. The details of our revisions are outlined in the point-by-point responses below.

Reviewer #1 (Remarks to the Author):

The authors performed a considerable work to improve their manuscript, and I appreciated the efforts performed. I think the present version is much clearer than the previous ones.

Thank you very much.

I only have a few very minor suggestions to further improve clarity
I think "characteristic / non characteristic species" should be clearly defined at the beginning of the paper

We have now provided a more detailed definition:

This algorithm simultaneously identifies modules of highly connected grid cells and species in an integrated approach^{19,25}. Grid cells within a module define a biogeographical region¹⁹ (Extended Data Figs. 2-8). Species assigned to the same module are considered characteristic of that biogeographical region¹⁹, meaning their distribution range is largely confined to that region, contributing to its unique biotic identity. In contrast, species present in a bioregion's grid cells but not grouped in the same module are deemed non-characteristic¹⁹—either because they are more strongly associated with another region or display an even distribution that precludes a clear regional association. See graphical description in Extended Fig. 1.

(Lines 85-92)

L 257: perhaps a question mark is missing here

We have added the missing question mark. Thank you.

L 283: to be honest, in most of cases pool size is more important than sorting, and in several cases pool size is much more important than sorting (eg for amphibians and reptiles you cannot write that the effect of sorting is comparable to the one of pool). Please re-shape the following sentences accordingly

We have adjusted the sentence to better reflect the relative importance of species sorting and pool size:

Linear regressions and variance partitioning showed that the influence of species sorting can be comparable to that of regional species pool size in some cases (dragonflies and mammals) or even greater in others (rays).

(Lines 283-285)

L 319: biodiversity within biogeographical regions?

We apologize for the ambiguity. By "global biodiversity patterns" we refer to our findings that highlight the importance of sorting characteristic and non-characteristic species within biogeographical regions for understanding global variations in species richness (Fig. 5). To ensure clarity, we have now explicitly stated:

*Furthermore, our core-to-transition hypothesis and results show that global variations in species richness can be better understood by unravelling the genesis of regional hotspots and the subsequent filtering of species to the rest of the biogeographical region.
(Lines 317-320).*

Reviewer #4 (Remarks to the Author):

First of all, I would like to thank the authors for having addressed all of my comments. I think that robustness analysis regarding the different data sources used are satisfactory.

Thank you very much

However, with respect to the method used to define biogeographical regions, the answer that I have received is not so satisfactory. My feeling is that you have not fully addressed my concern. What I asked was, if I were to use a method that is not 'specifically' for community detection but considers other connectivity patterns (including the modular one you are using), will I find the same patterns that you find?

The answer that you have given me is, in short, that you do not think these other approaches are appropriate, but I disagree. I fully understand that there is literature that uses community detection algorithms to define biogeographical regions (Vialha and Antonelli 2015 and other references). However, that does not mean that this is the best way to model the large scale organization of the network of species abundance in habitats. In my book, the fact that other people do it like this does not mean I should. If there is a possibility that other patterns could explain the data it would still be an interesting finding worth talking about, hence my honest concern. I believe, that I asked a very direct question that is relevant to your findings, and the only way to answer it is to run the stochastic block model based algorithm I suggested using the software developed by T Peixoto (which by the way has been demonstrated theoretically to lower the resolution limit if you use a hierarchical prior) and see what you find. Scientifically, this is the right thing to do because I do not think that you can argue that methods that intrinsically consider bipartite graphs and the best way to summarize the information in those graphs are a priori worse than the approach that you are using.

In all honesty, I know that you have done a lot of work but I also think that you need to fully address all of my concerns before I can accept. Making this very simple experiment and explaining the results are truly going to be good for the ecological community in general. As a final point, these approaches have also been used by ecologists in the analysis of other types of ecological data successfully (for instance see the work of S Kéfi and collaborators) so it is not foreign to the community.

We apologize for any misunderstanding regarding our rationale for selecting this analytical approach. We recognize that our previous response may not have sufficiently demonstrated the

robustness and reliability of our bioregion delineation. To address this, we have conducted the requested sensitivity analyses using the Stochastic Block Model (SBM) proposed by Peixoto to delineate bioregions. Our results show strong congruence with our main findings. Both Infomap and SBM provide multiple hierarchical levels (for example, realms, provinces, or districts). Using the Infomap partition and hierarchical level presented in our main text as a reference, we identified the most comparable hierarchical level in the SBM output. The similarity, measured by the Adjusted Mutual Information (AMI), was consistently high across all taxa ($AMI = 0.75 \pm 0.03$; mean \pm standard error). We have incorporated these results into our revised manuscript and Supporting Information Appendix B.

In addition to these new sensitivity analyses comparing Infomap-based and SBM-based bioregion delineations, we have clarified in our manuscript that previous studies have shown that Infomap produces results comparable to, or even superior to, well-established methods in biogeography such as agglomerative hierarchical clustering and modularity-based approaches (Vilhena and Antonelli 2015; Castello et al. 2017; Bloomfield et al. 2018). Infomap relies on information and compression theory, detecting clusters by minimizing the description length of information flow in a network (Rosvall & Bergstrom 2008). In contrast, SBMs use probabilistic inference, identifying groups by estimating the likelihood of connections between nodes (Peixoto 2017). Modularity-based methods focus on network topology, detecting clusters by maximizing internal connectivity relative to a null model (Newman & Girvan 2004). Agglomerative hierarchical clustering follows a sequential merging approach, grouping locations into a tree-like structure (Sokal and Sneath 1963; Jain et al. 1999) based on predefined similarity matrices such as beta-similarity. Determining the optimal number of clusters requires an external criterion, such as the "knee" method (Vilhena & Antonelli 2015) or SIMPROF permutation testing (Costello et al. 2017).

Despite these methodological differences, bioregions detected by Infomap remain largely similar with those delineated by other contrasting methods (Vilhena and Antonelli 2015; Castello et al. 2017; Bloomfield et al. 2018). For example, Bloomfield et al (2018) demonstrated the similarity in bioregion delineation when modeling *Acacia* sp. distribution ranges in Australia using agglomerative hierarchical clustering, Infomap, and modularity (we include a copy of their Fig. 1 below).

Fig. 1 from Bloomfield et al. (2018). Analyses of *Acacia* sp. dataset comparing three contrasting bioregionalization methods. Panel A shows the result of the clustering and dendrogram based on beta

similarity in grid cells. Panel B shows the results of the map equation (Infomap). Panel C shows the results of modularity analysis.

Even when multiple methods yield similar results, the absence of ground truth for comparison—like in biogeographical studies—can introduce uncertainty regarding the validity and realism of delineated bioregions. Biogeographical regions are defined as areas of the Earth's surface characterized by distinct species pools (Sclater 1858; Wallace 1876). Therefore, if the identified bioregions accurately reflect these distinctions, species characterizing different bioregions should exhibit minimal overlap in their distribution ranges, whereas species within the same bioregion should show substantial overlap. Vilhena and Antonelli (2015) provided empirical support for this expectation by analyzing global amphibian distribution data using Infomap—the same analytical tool and dataset employed in our study (see their Fig. 3 below). Their findings reinforce that Infomap-based bioregions are not only consistent with those identified by alternative methods (as demonstrated above) but also align with fundamental biogeographical principles and expectations of what constitutes a biogeographical region.

Figure 3 of Vilhena and Antonelli (2015). [Original caption]. Results from the network analyses for the world's amphibians. (a) Amphibian biogeographical regions of the world determined from geographical range data. Similar colours indicate membership to a higher level clustering, in this case equivalent to realms. The analysis used a resolution of two degree grid cells. (b) Species range limits coloured by region. Geographically close and neighbouring regions were given contrasting colours to highlight boundaries and boundary mixing. Each geographical range polygon was plotted with a low opacity (0.1), from largest to smallest on a global level, so that regions with more species appear brighter. (N=6,100 species).

We acknowledge that evaluating a method's performance in the absence of a ground truth, as discussed above, is not ideal. The most rigorous way to assess performance is through

comparisons where the expected output is already known, as is typically done using synthetic data. While this specific issue lies beyond the scope of our study and falls within the domain of methodological research, we have now explicitly stated in our manuscript that multiple studies using synthetic datasets with known ground truth consistently rank Infomap among the top-performing methods (Lancichinetti & Fortunato, 2009; Aldecoa & Marín, 2013; Tandon et al., 2021).

It is not the aim of our study to determine whether Infomap is the best tool for delineating biogeographical regions or for community detection. However, we hope that our new analyses, together with reference to previous literature, reinforce the reliability of Infomap-based bioregions. More importantly, beyond delineating biogeographical regions, our primary objective is to identify their characteristic species to describe biodiversity patterns. Since biogeographical regions are distinct areas of the Earth defined by the presence of unique species pools, we require a one-to-one correspondence between bioregions and species pools (i.e., one species pool per bioregion). While other methods have useful applications, we selected Infomap because it has been demonstrated to robustly identify this one-to-one correspondence between species pools and regions (Vilhena & Antonelli, 2015; Bernardo-Madrid et al., 2019; Edler et al., 2017). We have clarified these points in the main text (Lines 482 to 511).

In summary, our findings are strongly supported by (i) the new sensitivity analyses comparing Infomap-based and SBM-based bioregions, (ii) previous literature demonstrating the consistency of Infomap-based bioregions with other widely used and accepted methods in biogeography, (iii) the high spatial congruence between bioregion boundaries and the distribution ranges of characteristic species, and (iv) the strong performance of Infomap in synthetic data evaluations. We have now made all these points explicit in the text. Together, these lines of evidence reinforce the validity of our biogeographical regions and their suitability for broader applications, including our primary objective: describing biodiversity patterns within bioregions.

We acknowledge that this information was either absent or not clearly stated in our previous version, despite its crucial importance. For this reason, we sincerely thank the reviewer for their insightful comments, which have helped improve our study and strengthen the clarity and robustness of our results and inferences.

References

- Aldecoa, R., & Marín, I. (2013). Exploring the limits of community detection strategies in complex networks. *Scientific reports*, 3(1), 2216.
- Bloomfield, N. J., Knerr, N., & Encinas-Viso, F. (2018). A comparison of network and clustering methods to detect biogeographical regions. *Ecography*, 41(1), 1-10.
- Costello, M. J., Tsai, P., Wong, P. S., Cheung, A. K. L., Basher, Z., & Chaudhary, C. (2017). Marine biogeographic realms and species endemism. *Nature communications*, 8(1), 1057.
- Edler, D., Guedes, T., Zizka, A., Rosvall, M., & Antonelli, A. (2017). Infomap bioregions: Interactive mapping of biogeographical regions from species distributions. *Systematic biology*, 66(2), 197-204.
- Jain, A. K., Murty, M. N., & Flynn, P. J. (1999). Data clustering: A review. *ACM Computing Surveys*, 31(3), 264–323.
- Lancichinetti, A., & Fortunato, S. (2009). Community detection algorithms: a comparative analysis. *Physical Review E—Statistical, Nonlinear, and Soft Matter Physics*, 80(5), 056117.
- Newman, M. E. J., & Girvan, M. (2004). Finding and evaluating community structure in networks. *Physical Review E*, 69(2), 026113.

- Peixoto, T. P. (2017). Bayesian stochastic block modeling. *Advances in Network Clustering and Blockmodeling*, Wiley.
- Rosvall, M., & Bergstrom, C. T. (2008). Maps of random walks on complex networks reveal community structure. *Proceedings of the National Academy of Sciences*, 105(4), 1118–1123.
- Sclater, P. L. On the general Geographical Distribution of the Members of the Class Aves. *Journal of the Proceedings of the Linnean Society of London. Zoology* 2, 130–136 (1858).
- Sokal, R. R., & Sneath, P. H. A. (1963). *Principles of Numerical Taxonomy*. W.H. Freeman.
- Tandon, A., Albeshri, A., Thayanathan, V., Alhalabi, W., Radicchi, F., & Fortunato, S. (2021). Community detection in networks using graph embeddings. *Physical Review E*, 103(2), 022316.
- Wallace, A. R. *The Geographical Distribution of Animals*. (Cambridge University Press, 1876). doi:10.1017/CBO9781139097109.